# Neutrophil Spatiotemporal Regulatory Networks: Dual Roles in Tumor Growth Regulation and Metastasis

**DOI:** 10.3390/biomedicines13061473

**Published:** 2025-06-14

**Authors:** Pengcheng Li, Feimu Fan, Bixiang Zhang, Chaoyi Yuan, Huifang Liang

**Affiliations:** 1Hepatic Surgery Centre, Tongji Hospital, Tongji Medical College, Huazhong University of Science and Technology, Wuhan 430030, China; pcli9609@163.com (P.L.); feimu_fan@foxmail.com (F.F.); bixiangzhang@hust.edu.cn (B.Z.); 2Hubei Key Laboratory of Hepato-Pancreato-Biliary Diseases, Wuhan 430030, China; 3Key Laboratory of Organ Transplantation, Ministry of Education, NHC Key Laboratory of Organ Transplantation, Key Laboratory of Organ Transplantation, Chinese Academy of Medical Sciences, Wuhan 430030, China

**Keywords:** neutrophil, tumor microenvironment, neutrophil extracellular traps, tumor metastasis, tumor dormancy

## Abstract

Neutrophils, accounting for 50–70% of circulating leukocytes, exhibit remarkable plasticity in tumor biology. Depending on tumor type and microenvironmental cues, they can exert either anti-tumor or pro-tumor effects. During tumor initiation, neutrophils exposed to chronic inflammation secrete cytokines and oncogenic microRNAs that promote genomic instability and malignant transformation. In tumor progression, neutrophils adopt context-dependent phenotypes and execute diverse functions, including polarization into anti-tumor (N1) or pro-tumor (N2) subsets; secretion of inflammatory and angiogenic mediators; formation of neutrophil extracellular traps (NETs); production of reactive oxygen and nitrogen species (e.g., H_2_O_2_ and nitric oxide); and modulation of immune cell infiltration and function within the tumor microenvironment. During metastasis, neutrophils facilitate cancer dissemination through three principal mechanisms: (1) promoting epithelial–mesenchymal transition (EMT) via inflammatory signaling, adhesion molecule interactions, and lipid metabolic support; (2) establishing pre-metastatic niches by remodeling distant organ stroma through NETs and matrix metalloproteinases; and (3) reactivating dormant tumor cells in response to chronic inflammation, viral infection, or stress hormones. Collectively, neutrophils function as central regulators across all stages of tumor evolution, influencing cancer growth, immune evasion, and metastatic progression. This review aims to provide a comprehensive synthesis of neutrophil-mediated mechanisms in the tumor microenvironment and highlight emerging strategies for neutrophil-targeted cancer therapy.

## 1. Introduction

Neutrophils, the most prevalent innate immune cells in bone marrow and peripheral blood, are widely acknowledged as a vital element of the immune system [1]. They utilize a range of strategies, such as phagocytosis, NETs formation and degranulation to efficiently eradicate pathogens [2]. Under physiological environments, neutrophils exhibit limited longevity, with their stimulation and trafficking tightly regulated to prevent potential damage to healthy tissues due to their highly cytotoxic responses [3]. Neutrophils manifest a dual nature in disease contexts, serving as both essential defenders and potential contributors to pathological processes. On one hand, they are essential in host defense by combating various pathogens [4,5,6] and facilitate tissue regeneration and angiogenesis post-injury by clearing apoptotic cell debris [7]. On the other hand, in diverse pathological conditions, such as cardiovascular disorders, autoimmune diseases, infectious diseases, allergic conditions, and vasculitides, the presence of both pro-inflammatory and anti-inflammatory neutrophil subpopulations has been observed, underscoring their functional complexity and heterogeneity [8,9,10,11,12,13,14,15,16].

The tumor microenvironment (TME) is populated by heterogeneous immune cell subsets, notably neutrophils, macrophages, and T cells, each contributing fundamentally to tumor immunobiology [17,18,19,20]. Neutrophils represent a quantitatively and functionally significant population within the immune landscape of most solid tumors [21,22,23,24]. As the predominant immune cell population, neutrophils exhibit complex and significant functions in the context of cancer. The evolving understanding of neutrophil heterogeneity and plasticity has revitalized their study in cancer pathogenesis. Emerging evidence indicates that neutrophils are far from the homogeneous population once presumed; rather, their functionality is profoundly influenced by the surrounding microenvironment, resulting in varying degrees of pro-tumorigenic or anti-tumorigenic properties [25,26]. Extensive research has revealed the pleiotropic anti-tumor capacities of neutrophils, including direct tumoricidal activity and the capacity to suppress metastasis [27,28,29,30]. Conversely, other investigations have revealed that tumor-associated neutrophils (TANs) can promote tumor progression by facilitating angiogenic switching and enhancing tumor cell motility, migration, and invasion [31,32] (Figure 1). Epidemiological studies further underscore a significant correlation between neutrophils and clinical outcomes across various tumor types [33,34,35,36,37,38].

This review synthesizes current knowledge on neutrophil ontogeny, functional plasticity, and their dichotomous roles in tumor progression. By delineating neutrophil-microenvironment crosstalk, we aim to identify actionable targets for intercepting cancer progression and metastasis.

## 2. Neutrophil Biology: Development, Homeostasis, and Circulation

### 2.1. Neutrophil Development and Functional Maturation

Under homeostatic conditions, neutrophils, as the most predominant immune effector cells in humans, constitute approximately 50–70% of the total peripheral white blood cell count. Their daily production exceeds 10^11^ cells [39], yet mature neutrophils lose their proliferative capacity. To sustain this high output and rapid turnover, the bone marrow dedicates about 65% of its hematopoietic space to granulocyte-monocyte lineage development under physiological conditions [40]. Notably, the conventional identification of granulocyte precursors primarily relies on histological observation via Giemsa staining after density gradient centrifugation [41]. This staging system, based on morphological features (including cell volume, nuclear-to-cytoplasmic ratio, chromatin condensation, and specific granule content), still suffers from limitations due to subjective interpretation.

Developmental studies indicate neutrophils arise from lymphoid-primed multipotent progenitors (LMPPs) during hematopoietic differentiation, with LMPPs themselves being derived from multipotent progenitors (MPPs) [42]. These progenitors undergo stepwise differentiation through the granulocyte–monocyte progenitor (GMP) stage, a process regulated by a cascade of key transcription factors [40,43,44]. Terminal differentiation follows a strict morphological progression: myeloblast → promyelocyte → myelocyte → metamyelocyte → band cell → mature neutrophil [40,45,46,47]. Importantly, the granule assembly cascade initiates with primary (azurophilic) granule formation at the promyelocyte stage, followed by robust production of secondary (specific) granules in myelocytes, culminating in tertiary (gelatinase) granule generation during terminal neutrophil maturation [40,48]. These subcellular structures compartmentalize effector molecules such as myeloperoxidase, defensins, antimicrobial peptides, as well as proteolytic enzymes like elastase and matrix metalloproteinases (MMPs), playing dual roles in antimicrobial defense and tissue homeostasis regulation [48,49] (Figure 2).

Following egress from the bone marrow, terminally differentiated neutrophils enter the circulatory compartment as mature circulating neutrophils, with species-dependent half-lives: ~7 h in humans [50,51] and 8–10 h in mice [52]. After migrating to inflammatory sites, their lifespan can extend to 24–48 h before spontaneous apoptosis [53], with certain microenvironments further delaying apoptosis [48,52,54].

### 2.2. Bone Marrow Reservoir and Regulatory Functions

As the most numerically dominant cell type in the bone marrow, neutrophils constitute the largest granulocyte reservoir in the body. Cross-species classical studies tracking their migration and distribution dynamics have revealed that the bone marrow-resident granulocyte pool exhibits a magnitude-order quantitative superiority over the circulating granulocyte pool [55,56,57].

Neutrophils are distinct from other immune cells as they are directly released into the peripheral circulation in a terminally differentiated mature state. The bone marrow not only regulates the homeostatic maturation of granulocytic precursors (spanning from myeloblasts to band-stage cells) but also serves as the final destination for peripheral circulating neutrophils. Approximately 30% of circulating neutrophils can re-enter the bone marrow via the CXCR4/CXCL12 axis-mediated homing mechanism, a process exhibiting pronounced 24-h circadian fluctuations [58,59].

Bone marrow-resident neutrophils exhibit unique immunomodulatory functions: under genotoxic stress, they secrete TNF-α to promote sinusoidal endothelial cell proliferation, thereby maintaining hematopoietic microenvironment homeostasis [60]. Adrenergic stimulation induces neutrophils to biosynthesize prostaglandin E2 (PGE2), which activates osteoblast function through prostaglandin E Receptor 4 (EP4) receptor-dependent mechanisms, consequently enhancing hematopoietic stem cell (HSC) homing and retention capabilities within the bone marrow microenvironment [61]. Furthermore, histidine decarboxylase-positive (HDC^+^) neutrophil subsets preserve quiescence and enhance regenerative potential in myeloid-biased HSCs [62].

Multiple molecular mechanisms regulate neutrophil residence in the bone marrow through a complex regulatory network: (1) Chemokine balance system: osteoblasts constitutively secrete CXCL12, anchoring mature granulocytes to the endosteal region via CXCR4, while endothelial cells and megakaryocytes produce CXCL1/2 to drive granulocyte mobilization through CXCR2 signaling [59,63,64,65,66,67]; (2) Adhesion molecule interactions: the Very Late Antigen-4-Vascular Cell Adhesion Molecule 1 (VLA-4-VCAM1) ligand-receptor pairing synergizes with proteases to reinforce neutrophil-stromal adhesion [68,69,70]; (3) Growth factor regulation: G-CSF disrupts retention equilibrium through triple mechanisms—promoting CXCR2 ligand (CXCL1 and CXCL2) expression in megakaryocytes through Thrombopoietin (THPO)-dependent pathways, suppressing CXCL12 synthesis in stromal cells, and downregulating CXCR4 receptor levels on neutrophils [67,71,72,73].

### 2.3. Circulatory Dynamics and Homeostatic Regulation

Under physiological homeostasis, neutrophils adhere to strict dynamic equilibrium: following release from the bone marrow, they undergo an approximately 12 h circulatory cycle (murine models indicate a half-life of ~8–10 h), with clearance ultimately achieved via tissue infiltration [58,74,75]. Notably, despite their transient residence in the bloodstream, their phenotypic alterations exhibit circadian rhythm-dependent aging characteristics, a phenomenon termed neutrophil aging [76]. Quantitative studies confirm that under steady-state conditions, the circulating pool constitutes merely 1–2% of the total systemic neutrophil population [77].

The maintenance of neutrophil homeostasis relies on an intricate multi-layered regulatory network external to the bone marrow: (1) IL-23/IL-17/G-CSF cascade regulation: Tissue-resident phagocytes (e.g., macrophages) control IL-17 production by γδ T cells, αβ T cells, and native lymphoid cells through IL-23 [78,79]. IL-17 drives bone marrow neutrophil release by inducing G-CSF production [80,81]. The timely clearance of apoptotic neutrophils (mediated by macrophages and dendritic cells) suppresses IL-23/IL-17 axis activity, forming a negative feedback loop [82,83,84,85]. (2) Gut microenvironment regulation: Gut commensal microbiota stimulate intestinal epithelial cells to secrete CXCL5, which promotes granulopoiesis through IL-17 induction [86,87]. (3) IL-1β as an inflammatory amplifier: IL-1β released during cellular damage or via inflammasome activation potently enhances IL-17 transcription, thereby amplifying G-CSF signaling [88,89].

## 3. Neutrophils as Key Mediators of Inflammation-to-Carcinogenesis

During the acute phase of bacterial or viral infections, neutrophils, as the vanguard of innate immunity, play a pivotal protective role. However, as the acute phase transitions to the chronic phase, neutrophils may engage in deleterious actions, promoting tumor formation and progression. The inflammation–cancer axis, first conceptualized by Virchow in 1863, now represents one of the hallmarks of cancer pathogenesis. In cellular pathology, he first described leukocyte infiltration within tumor microenvironments and pioneered the concept that chronic inflammation constitutes a critical soil for carcinogenesis [90,91]. Epidemiological studies reveal that ~25% of malignancies are directly linked to chronic inflammation, with pathological subtypes spanning multiple anatomical systems: gastrointestinal (esophagitis → esophageal cancer, colitis → colorectal cancer, pancreatitis → pancreatic cancer), hepatobiliary (hepatitis → hepatocellular carcinoma), respiratory (pneumonitis → lung cancer) [90,92,93,94,95,96,97,98]. Modern molecular pathology confirms that pathogen-associated molecular patterns (PAMPs) stimulate Toll-like receptor (TLR)-dependent NF-κB activation, establishing a pro-tumorigenic microenvironment through the continuous release of inflammatory cytokines such as IL-6 and TNF-α by neutrophils, thus providing the molecular basis for the transformation from inflammation to tumorigenesis [99]. Pathological analyses reveal that activated infiltrating neutrophils in inflammatory bowel disease (IBD) patients and experimental colon injury models secrete pro-inflammatory microvesicles enriched with oncogenic molecules (e.g., miR-23a/miR-155), which trigger genomic instability by suppressing DNA repair pathway genes [100]. Strikingly, miR-155 directly mediates neutrophil-associated DNA double-strand breaks and repair network dysregulation under acute colonic injury, driving colorectal carcinogenesis and malignant progression [101].

Reactive oxygen species (ROS), particularly those derived from neutrophils, play a pivotal role in chronic inflammatory lesions by inducing carcinogenic mutations, with colitis serving as a representative example [102,103]. Helicobacter pylori infection and IBD induce extensive neutrophil infiltration and oxidative burst, generating ROS and RNS, including superoxide (O_2_˙^−^), H_2_O_2_, hypochlorous acid (HOCl), hydroxyl radicals (˙OH), and peroxynitrite, which drive gastric and colorectal carcinogenesis. Produced primarily via NADPH oxidase and mitochondrial electron transport chain, ROS/RNSs, while intended to eliminate pathogens, cause widespread host cell damage, including DNA lesions, lipid peroxidation, and protein oxidation. These insults provoke genomic instability, membrane dysfunction, and protein impairment, compromising cellular integrity. Additionally, ROSs act as second messengers, activating NF-κB, AP-1, and matrix metalloproteinase-1 (MMP1), upregulating IL-8, VEGF, and iNOS, thereby promoting inflammation, proliferation, survival, angiogenesis, and tumor invasion/metastasis. Oxidative stress further induces epigenetic silencing of tumor suppressor genes (e.g., p53, Rb, hMLH1, p16, p14) via DNA methylation. Collectively, neutrophil-driven oxidative stress significantly contributes to H. pylori-associated gastric cancer and IBD-related colorectal cancer by damaging cellular components, modulating signaling pathways, and inducing genetic/epigenetic aberrations.

Inflammatory responses act as key tumorigenic drivers by disrupting tissue homeostasis [94], with neutrophils functioning as central effector cells. Within the TME, soluble mediators and intercellular interactions form dynamic regulatory networks, where neutrophils act as hub nodes, constructing molecular bridges linking inflammatory processes to oncogenic pathways.

## 4. The Behavior of Neutrophils in the Tumor Progression and Tumor Metastasis

Neutrophils exhibit dual regulatory functions within the TME, actively participating in the dynamic regulation of metastatic cascades through either pro-tumorigenic or anti-tumorigenic mechanisms [104,105]. Their functional plasticity allows them to respond to a wide array of inflammatory signals and TME-derived factors, thereby exerting contrasting effects on tumor progression [106]. A clear understanding of these dual roles is essential for the development of effective neutrophil-targeted cancer therapies (see Figure 3 and Table 1). Furthermore, neutrophils play a pivotal role in promoting tumor cell migration, facilitating the formation of pre-metastatic niches, and reactivating dormant tumor cells at distant sites (see Figure 4 and Table 2).

### 4.1. The Dualistic Behavior of Neutrophils in the Tumor Progression

#### 4.1.1. Pro-Tumorigenic Roles of Neutrophils

##### Pro-Angiogenesis

Neutrophils, particularly those polarized toward the N2 phenotype, are key contributors to tumor angiogenesis through the secretion of multiple pro-angiogenic factors, including FGF2, VEGF-A, ANGPT1, CXCL8, HGF, and MMP9 [36,104,105,107,108,109,110,111,112,113,114,115,116,117], which collectively facilitate tumor vascularization and metastatic spread. Studies have shown that metastasis-associated neutrophils exhibit markedly elevated FGF2 expression relative to their naïve counterparts, contributing to enhanced hepatic metastatic progression through increased vascular density and branching [107]. In addition, MMP9 secreted by these neutrophils further enhances tumor neoangiogenesis, thereby promoting the development of pancreatic and lung cancers [108]. Importantly, research has demonstrated that MMP9 released by bone marrow-derived neutrophils collaborates with other hematopoietic lineage cells, possibly by modulating the tumor microenvironment or extracellular matrix remodeling, to promote the initiation of squamous cell carcinoma [118].

##### NETs Formation

NETs promote tumor metastasis through both physical entrapment and molecular signaling mechanisms. Within their extracellular fibrous network, NETs capture circulating tumor cells (CTCs), shielding them from the cytotoxic effects of CD8+ T cells and NK cells while simultaneously facilitating metastatic adhesion [104,119,120]. Higher levels of circulating NETs have been shown to correlate with increased metastatic burden in advanced esophageal cancer, gastric cancer, and lung cancer [120]. Amyloid-beta, secreted by CAFs, promotes NETs formation and contributes to tumor progression [121]. At the mechanistic level, tumor cells utilize the CCDC25 receptor to recognize DNA associated with NETs, thereby activating the ILK-β-parvin signaling axis and enhancing the adhesion, motility, and proliferation of liver-metastatic cells [122]. This process accelerates the growth of liver metastases, providing direct evidence for the pro-metastatic role of NETs. Studies in mouse models further demonstrate that NETs infiltrating into peritumoral tissues can enhance metastatic potential [123,124]. Furthermore, inflammatory signals not directly linked to tumor initiation can also trigger neutrophil NETosis [125,126].

##### Immunosuppressive Microenvironment Remodeling

The suppression of T cell activity by neutrophils represents a critical mechanism underlying tumor immune escape, with distinct mechanisms observed in different tumor types: in colorectal cancer, N2-polarized neutrophils promote immune evasion by activating latent TGFβ via MMP9, thereby suppressing the function of tumor-infiltrating T cells [127]; in ovarian cancer, these TANs mediate IL-8-dependent immunosuppression (suppressing CD8^+^ T cells) through the Jagged2 signaling pathway [128]. Furthermore, tumor-derived small extracellular vesicles (sEVs) deliver serine protease inhibitors (serpins), which reprogram neutrophils toward a pro-tumorigenic phenotype characterized by an extended lifespan and an immunosuppressive expression pattern. These neutrophils functionally enhance tumor cell epithelial–mesenchymal transition and potently inhibit cytotoxic CD8+ T cells, thereby reducing their tumor-killing efficacy [129]. In the tumor microenvironment, IL-1β induces G-CSF production, promoting both neutrophil expansion and functional polarization. Tumor-educated neutrophils gain the capacity to suppress CD8+ T cell cytotoxicity, thereby facilitating tumor progression [130,131,132,133,134].

#### 4.1.2. Anti-Tumorigenic Roles of Neutrophils

##### Oxidative Stress-Mediated Tumor Cell Killing

Neutrophils exert dual antitumor effects through a synergistic mechanism involving ROS and RNS. Mechanistic studies have demonstrated that H_2_O_2_ secreted by neutrophils activates the TRPM2 calcium channel on tumor cell membranes, leading to a rapid increase in intracellular Ca^2+^ levels. This elevation in Ca^2+^ subsequently initiates calpain-mediated activation of caspase-3, culminating in apoptotic cell death [135,136]. In addition, neutrophil-derived H_2_O_2_ has been shown to suppress pulmonary metastatic colonization [30]. Moreover, research by the Finisguerra group revealed that HGF interacts with the MET receptor on neutrophil surfaces, thereby activating the iNOS signaling axis and promoting NO production, which enhances tumor cell cytotoxicity [137]. Moreover, N1-polarized neutrophils potentiate anti-tumor immune responses and improve therapeutic outcomes via oxidative stress-mediated suppression of IL-17+ γδ T cells [138].

##### Protease-Dependent Selective Cytotoxicity

NE demonstrates a distinct ability to target tumors and has been shown to selectively destroy cancer cells while inhibiting tumorigenesis [139]. This targeted cytotoxic effect works alongside oxidative mechanisms, thereby enhancing the overall antitumor response.

**Figure 3 biomedicines-13-01473-f003:**
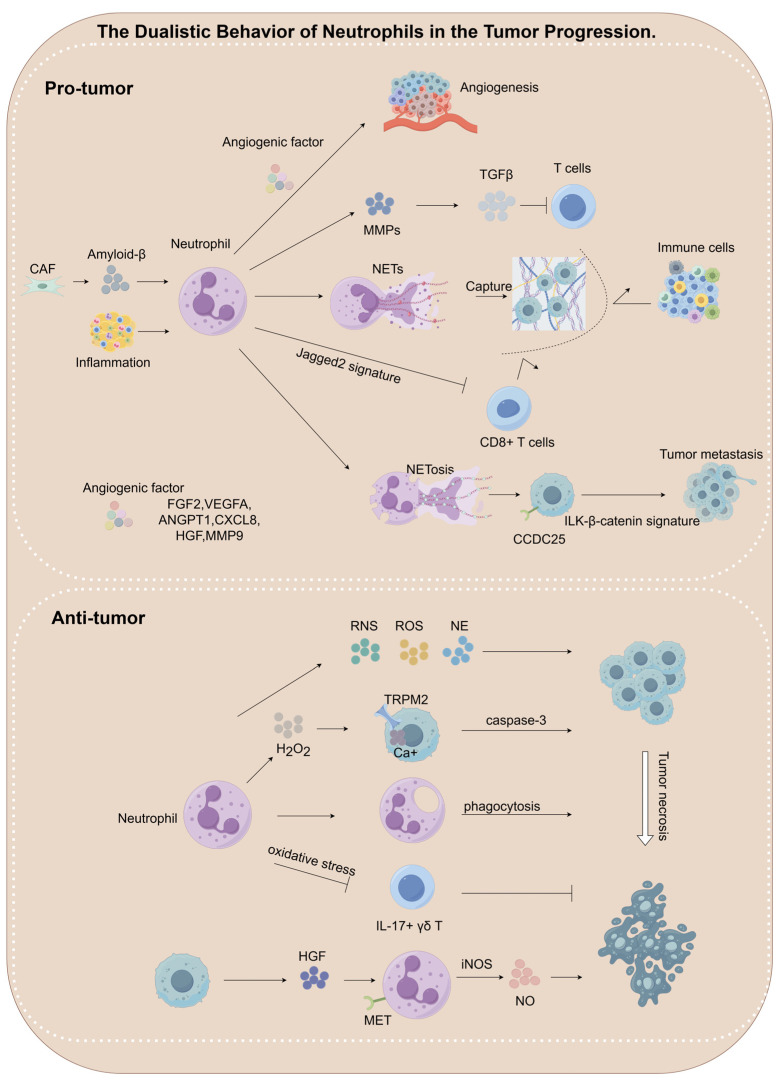
The dualistic behavior of neutrophils in the tumor progression. Neutrophils exhibit functional plasticity and play a dual role in modulating tumor progression, which can be either tumor-promoting or tumor-suppressive, depending on cues from the microenvironment and epigenetic regulation. Tumor-promoting neutrophils contribute to cancer progression through multiple pathways, including the secretion of pro-angiogenic factors (e.g., VEGF, MMP9, FGF2), induction of immunosuppression (e.g., via TGFβ and NETs), and suppression of immune cell activity. In contrast, tumor-suppressive neutrophils mediate cytotoxicity through multiple mechanisms, such as the generation of ROS, RNS, and NE.

**Table 1 biomedicines-13-01473-t001:** Mechanisms underlying the role of neutrophils in tumor growth regulation.

Characteristics	Mechanisms	References
Anti-tumor	Neutrophils suppress the proliferation of IL-17+ γδ T cells by secreting ROS, thereby exerting anti-tumor effects in melanoma and hepatocellular carcinoma.	[138]
	Neutrophils acquire anti-tumor phenotypes in breast cancer under the influence of ACE inhibitors and AGTR1 antagonists, thereby suppressing tumor growth.	[140]
	Neutrophils undergo N1-dominant reprogramming upon TGF-β blockade, resulting in effective tumor suppression across various cancers including breast, bile duct, colon, lung, and melanoma.	[141,142,143,144,145,146,147,148]
	Neutrophils secrete H_2_O_2_, which binds to the TRPM2 receptor on tumor cells, inducing Ca²^+^ influx and promoting tumor cell death through activation of the caspase-3 apoptosis signaling pathway in breast cancer.	[135,136]
	HGF/MET-dependent nitric oxide release by neutrophils promotes cancer cell killing in many tumors, including fibrosarcoma, colon cancer, lung cancer, melanoma, and hepatocellular carcinoma.	[137]
	Neutrophils inhibit metastatic seeding in the lungs by generating H_2_O_2_ in breast cancer, lung cancer, melanoma.	[30]
	Neutrophils release catalytically active NE, which hydrolyzes the CD95 death domain to selectively eliminate cancer cells in pan-cancers.	[139]
Pro-tumor	Neutrophils suppress tumor-infiltrating T cells in colon cancer via MMP9-mediated activation of TGFβ in colon cancer.	[127]
	Neutrophils contribute to skin carcinogenesis by releasing MMP9.	[118]
	CAFs secrete Amyloid β, which enhances the formation of NETs, thereby promoting tumor progression in melanoma, pancreatic cancer.	[121]
	Neutrophils promote angiogenesis through FGF2 secretion.	[107]
	Neutrophils promote angiogenesis through MMP9 secretion.	[36,108,117]
	IFN-β inhibits the production of VEGF and matrix MMP9 by neutrophils, consequently suppressing angiogenesis in melanoma.	[111]
	Neutrophils promote angiogenesis by releasing VEGF, HGF, and Angiopoietin-1.	[112,113,114,115,116]
	NETs promote tumor cell metastasis in esophageal, gastric, colon and lung cancer.	[120]
	Neutrophil-derived NETs engage the CCDC25 receptor on tumor cells, triggering ILK-β-parvin signaling and promoting metastasis in breast and colon cancer.	[122]
	NETs promote tumor cell metastasis in breast cancer.	[123]
	Lung inflammation promotes metastasis through neutrophil protease-mediated degradation of Tsp-1 in melanoma.	[125]

### 4.2. The Role of Neutrophils in Tumor Metastasis

#### 4.2.1. Facilitating Tumor Migration

##### Mechanisms of EMT Regulation

Neutrophils orchestrate tumor cell EMT by secreting mediators such as IL-17, TGF-β, and NE, which synergistically enhance transcriptional reprogramming [149,150,151]. Gastric cancer-derived HMGB1-enriched exosomes activate the TLR4/NF-κB pathway in neutrophils, triggering the autophagy-dependent secretion of IL-1β and OSM. This cascade markedly amplifies tumor cell migratory capacity [152].

##### Adhesive Support and Metastatic Niche Formation

Neutrophils facilitate metastatic colonization through both biomechanical interactions and molecular anchoring mechanisms. Clinical cohort analyses reveal a positive correlation between the proportion of circulating tumor cell–neutrophil clusters (CTC-Neu clusters) in peripheral blood and metastasis risk in breast cancer patients [153]. In murine melanoma models, IL-8-induced binding of neutrophil surface integrin β2 to tumor cell ICAM-1 stabilizes circulating tumor cells and potentiates their pulmonary dissemination [154]. Intravital microscopy further confirms that Lewis lung carcinoma cells form stable adhesions with neutrophils within hepatic microenvironments [155].

##### Metabolic Reprogramming and Energy Supply Networks

Neutrophils contribute to metastatic progression by remodeling lipid metabolism to provide energetic support for disseminating tumor cells [156]. In breast cancer lung metastasis models, angiopoietin-like 4 (ANGPTL4) secreted by pulmonary stromal cells impairs ATGL function in neutrophils, thereby promoting intracellular triglyceride storage. Through macropinocytosis-lysosomal pathways, single neutrophils transfer lipids to tumor cells, fulfilling their bioenergetic demands [156].

#### 4.2.2. Promotion of Pre-Metastatic Niche Formation

##### Theoretical Evolution of Pre-Metastatic Niches

Stephen Paget’s 1889 “Seed and Soil” hypothesis [157] has been validated at the molecular level in modern metastasis research. This theory emphasizes that metastatic dissemination requires dynamic interactions between CTCs and distal organ microenvironments: CTCs, as biologically active “seeds,” must locate a “permissive soil”—termed the PMN—characterized by enhanced vascular permeability, infiltrated immunosuppressive cells, and extracellular matrix remodeling. Kaplan et al. first demonstrated the spatiotemporal plasticity of PMN through organ-specific recruitment experiments involving bone marrow-derived cells [158,159].

##### Neutrophils as Central Drivers of PMN Formation

In the MMTV-PyMT transgenic mouse model of spontaneous breast cancer metastasis, CD11b^+^Ly6G^+^ neutrophils form sentinel clusters at target sites before tumor cell arrival, with their density showing a strong positive correlation with subsequent metastatic burden [160]. Therapeutic targeting of neutrophils with anti-Ly6G antibodies markedly attenuated metastatic colonization in the lungs [160]. The nicotine-exposure further corroborates that early neutrophil infiltration into the lungs elevates the incidence of melanoma lung metastasis [161].

##### Exosome-Mediated Organotropic Programming

Tumor-derived exosomes (TDEs) remodel distal organ microenvironments through molecular navigation mechanisms. The priming effect of TDEs on neutrophils was first demonstrated in melanoma [162], with subsequent studies showing that exosomes from diverse metastatic tumors specifically home to their typical metastatic organs, establishing organotropic metastatic tropism [163]. Further research reveals that lung-tropic TDEs activate TLR3 in alveolar epithelial cells, inducing chemokine production and recruitment of immunosuppressive neutrophils [164]. These pro-metastatic neutrophils, recruited to the lungs, enhance metastasis in breast cancer models by releasing MMPs to modify pulmonary stromal architecture [165].

##### Regulatory Networks of NETs

In the 4T1 breast cancer model, mitochondria DNA-enriched NETs released by senescent neutrophils promote collagen remodeling and accelerate metastatic colonization [166]. Clinicopathological analyses of advanced ovarian cancer patients demonstrate elevated H3Cit^+^MPO^+^ NETs density in omental tissues compared to early-stage cases, correlating significantly with reduced progression-free survival [124]. Another study highlights that NETs-driven PMN formation depends on complex interactions among diverse stromal cell populations within the tumor microenvironment [167].

#### 4.2.3. Activation of Dormant Tumor Cells

##### Biological Features of Tumor Cell Dormancy

Metastatic tumor cells can remain in a dormant state characterized by dynamic equilibrium between proliferation and apoptosis. This process involves impaired angiogenic mimicry and cell cycle arrest. Clinical cohort studies demonstrate that even after successful resection of primary tumors, patients may experience metastatic recurrence due to activation of dormant cells [168]. This dormant state is generally maintained until external stimuli from the microenvironment reignite proliferative activity [169].

##### NET-Mediated Reactivation of Dormant Cells

Chronic inflammation-induced NETs serve as critical drivers of dormant cell activation. NE and MMP9 released from NETs cleave adhesion proteins and activate integrin α3β1-dependent signaling, thereby reigniting dormant tumor cell proliferation [126]. Integrin signaling, particularly through integrin β1, has been implicated in regulating the switch from tumor dormancy to metastatic proliferation [169]. Of clinical relevance, data from breast cancer patients with COVID-19 comorbidity reveal a positive correlation between pulmonary NETs density and circulating tumor cell activation rates, suggesting that virus-associated pneumonia may accelerate tumor recurrence via NETs-mediated mechanisms [126,170,171].

##### The Regulatory Role of Secreted Protein

Using murine models—including Mmp9^−/−^ mice and anti-Ly6G antibody-mediated neutrophil depletion—experimental evidence reveals that 14,15-epoxyeicosatrienoic acid (14,15-EET) initiates a pro-metastatic cascade involving neutrophil-dependent MMP9 production, which both enhances angiogenesis and disrupts tumor dormancy [172].

##### Stress Hormone-Driven Activation

In ovarian and pulmonary neoplasms, stress hormone signaling activates neutrophils, which in turn facilitates the transition from dormant micrometastases to clinically detectable recurrences [173]. This activation involves a tripartite cascade: S100A8/A9 protein secretion, upregulated MPO activity, and oxidized lipid accumulation. These events converge to activate dormant tumor cells via the FGF signaling axis, facilitating their transition from quiescence to proliferative states [173].

**Figure 4 biomedicines-13-01473-f004:**
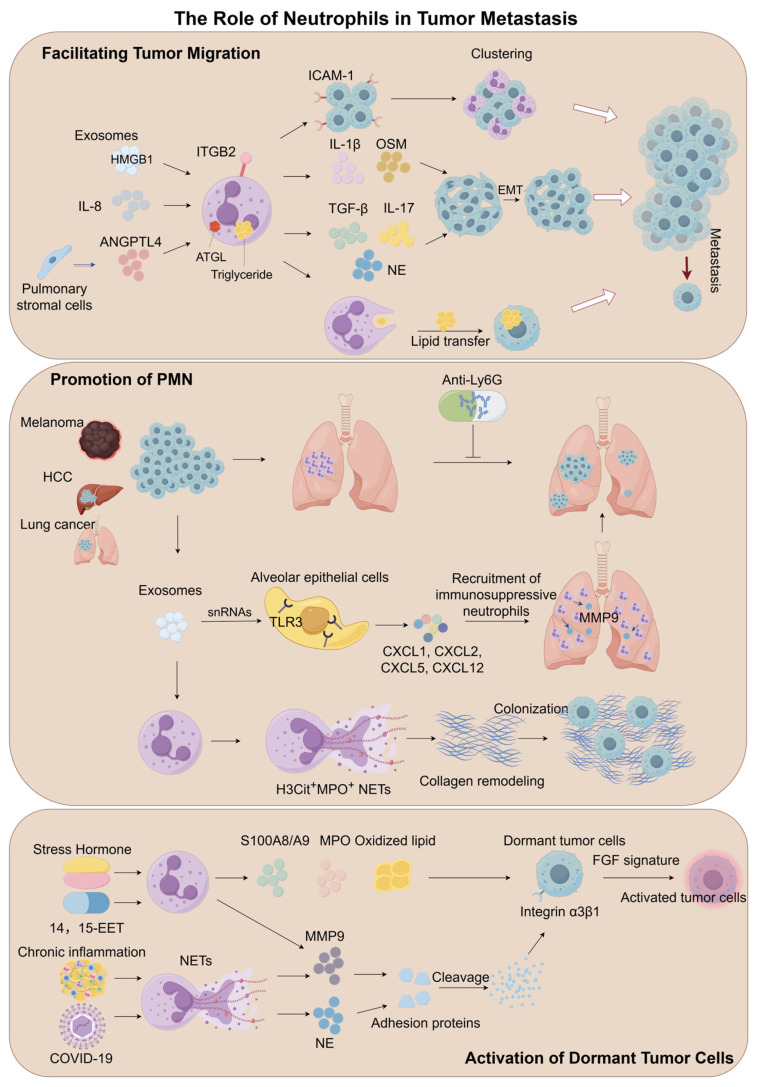
The role of neutrophils in tumor metastasis. Neutrophils drive tumor migration, PMN formation, and dormant tumor cell activation. They promote EMT via IL-17, IL-1β, OSM, TGF-β, and NE, enhance metastatic adhesion through CTC-Neu clusters, and support tumor energy needs via lipid transfer. Neutrophils form PMNs by remodeling stromal architecture and releasing MMPs and NETs, which also reactivate dormant cells by cleaving adhesion proteins and activating integrin signaling. Chronic inflammation, COVID-19, and 14,15-EET can also activate tumor cells. Tumor-derived exosomes and stress hormones further amplify neutrophil-driven metastasis, highlighting their critical role in cancer progression and recurrence.

**Table 2 biomedicines-13-01473-t002:** Mechanisms underlying the role of neutrophils in tumor metastasis.

Characteristics	Mechanisms	References
Facilitating tumor migration	TANs produce IL-17a, which promotes EMT of GC cells through JAK2/STAT3 signaling in gastric cancer.	[149]
	Neutrophils secrete NE, which cleaves E-cadherin on tumor cell surfaces while inducing nuclear translocation of β-catenin and Zeb1, promoting tumor cell EMT in pancreatic ductal adenocarcinoma.	[150]
	GC-Ex activates neutrophils through the HMGB1/TLR4/NF-κB signaling pathway, thereby promoting tumor metastasis in gastric cancer.	[152]
	Clinical cohort analysis shows that the proportion of circulating tumor cell-neutrophil clusters in the peripheral blood of breast cancer patients is positively correlated with metastasis risk in breast cancer.	[153]
	ICAM-1 on melanoma cells and β2 integrin on neutrophils interacted, promoting anchoring to vascular endothelium in melanoma.	[154]
	The adhesion of lipopolysaccharide-activated neutrophils to cancer cells was mediated by neutrophil Mac-1/ICAM-1 in lung cancer.	[155]
	Upon contact with MC cells, neutrophils experience ATGL suppression, leading to intracellular lipid accumulation, which is transferred to tumor cells via the macropinocytosis-lysosome pathway, promoting tumor metastasis in breast cancer.	[156]
	Neutrophils accumulate in the pre-metastatic lung microenvironment and promote tumor cell colonization by secreting leukotrienes in breast cancer.	[160]
Promotion of pre-metastatic niche formation	NET formation in rendering the PMN conducive for implantation of ovarian cancer cells, while PAD4 plays a critical role in NETs formation in ovarian cancer.	[124]
	Chronic nicotine exposure induces neutrophil recruitment in the lung, where neutrophils release LCN2, promoting MET in tumor cells, thereby enhancing their colonization and metastatic potential in breast cancer.	[161]
	Lung epithelial cells are critical for initiating neutrophil recruitment and lung metastatic niche formation by sensing tumor exosomal RNAs via TLR3 in melanoma.	[164]
	Exosomes from highly metastatic melanoma increased the metastatic behavior of primary tumors by permanently “educating” bone marrow progenitors via the MET receptor in melanoma.	[162]
	Neutrophils operate to facilitate extravasation of tumor cells through the secretion of IL1β and matrix metalloproteinases in breast cancer and melanoma.	[165]
	Neutrophils promote breast cancer lung metastasis through the SIRT1-Naged-NETs axis in breast cancer.	[166]
	LMSCs promote neutrophil recruitment and NETs formation by secreting C3, thereby facilitating cancer cell metastasis to the lung in breast cancer.	[167]
Dormancy activation	NE and MMP9 in NETs promote ECM remodeling, activating the integrin α3β1-FAK/ERK/MLCK/YAP signaling pathway to enhance dormant tumor cell proliferation in breast cancer.	[126]
	14,15-EET induces G-CSF/IL-6 production in vivo, enhancing STAT3 activation in neutrophils to promote MMP-9 expression and suppress TRAIL expression, with neutrophil-derived MMP-9 being essential for inducing angiogenesis in dormant micrometastases in melanoma.	[172]
	Stress hormones trigger release of S100A8/A9 proteins from neutrophils, which activate MPO, leading to oxidized lipid accumulation. These lipids, upon release, upregulate the fibroblast growth factor pathway in tumor cells, promoting their exit from dormancy and formation of new tumor lesions in lung cancer.	[173]

## 5. Single-Cell and Spatial Omics Uncover Neutrophil Heterogeneity and Therapeutic Targets

Single-cell and spatial omics have elucidated the precise localization and functional roles of neutrophils within the TME across various cancer types, highlighting their heterogeneity and interactions with other cellular components. In prostate cancer, integrated scRNA sequencing and high-resolution spatial transcriptomics revealed an immunosuppressive TME driven by myeloid cells, including neutrophils, and exhausted T cells, contributing to tumor recurrence and metastasis [174]. In non-small cell lung cancer (NSCLC), single-cell and spatial transcriptomic analyses underscored the pervasive presence of myeloid cells, including neutrophils, and their pivotal role in disease progression, providing a high-resolution molecular map of their contributions to the TME [175]. For pancreatic ductal adenocarcinoma (PDAC), multi-omics analyses have elucidated the heterogeneity of neutrophils in the PDAC tumor microenvironment at single-cell resolution, identifying four distinct neutrophil subsets and revealing their dynamic functions, metabolic reprogramming, and regulatory mechanisms [176]. Additionally, single-cell and spatial transcriptomic studies have characterized multicellular dynamics associated with neoadjuvant therapy in PDAC, providing a framework for understanding neutrophil contributions to the spatial TME, despite not directly addressing them [177]. In ovarian cancer, a microfluidic-based TIME-on-Chip model demonstrated that NETs promote collective tumor invasion, offering an in vitro platform to study neutrophil spatial functions [178]. In glioblastoma: single-cell RNA sequencing highlighted the emergence of pro-tumor myeloid-derived suppressor cells (MDSCs), including neutrophils, during disease progression [179]. In Gastric cancer, single-cell RNA sequencing post-PD-1 therapy identified increased circulating neutrophils, with the NE-1 subset exhibiting an activated phenotype (high MMP9, S100A8, S100A9, PORK2, TGF-β1 expression) and interacting with malignant epithelial cells and M2 macrophages via chemokine pathways to promote tumor progression [180]. Pan-cancer analyses across 17 cancer types and 225 samples revealed transcriptional heterogeneity among tumor-associated neutrophils, forming 10 distinct states; HLA-DR+CD74+ antigen-presenting neutrophils correlated with improved survival, while VEGFA+SPP1+ angiogenic neutrophils were associated with poorer prognosis. Interventions like leucine diets or delivery of antigen-presenting neutrophils enhanced anti-PD-1 immunotherapy in mouse models and human tumor fragment models, suggesting novel neutrophil-based therapeutic strategies [181]. Additionally, a single-cell tumor immune atlas from 217 patients across 13 cancers, integrated with the SPOTlight tool, enabled in situ immune population mapping, supporting patient stratification based on immune composition [182].

Spatial omics, integrated with single-cell omics, will advance our understanding of neutrophil heterogeneity, spatial distribution, and their complex interactions with tumor progression and therapeutic responses within the tumor microenvironment. These high-resolution molecular profiles, coupled with in vitro models simulating three-dimensional tumor microenvironments, will provide a robust foundation for developing precise neutrophil-targeted anticancer therapies.

## 6. Targeting Neutrophils in Cancer Therapy: Strategies and Approaches

TANs play a pivotal role in tumor progression. This section explores various strategies to target and inhibit TANs recruitment, as well as methods for monitoring and modulating neutrophil behavior to improve cancer treatment outcomes.

### 6.1. Strategies to Inhibit TANs Recruitment

The recruitment of TANs is a critical event in tumor progression, and targeting this process offers various strategies for cancer therapy. Chemokines and their receptors play an essential role in the migration of neutrophils to tumor sites. For example, CXCL2 promotes neutrophil recruitment and inflammatory responses via the CXCR2 axis, and inhibiting CXCR2 can alleviate myeloid inflammation and reverse treatment resistance in prostate cancer [183,184]. The ALX/FPR2 receptor regulates neutrophil infiltration based on ligands such as serum amyloid A, and targeting this receptor holds promise for controlling tumor-associated inflammation [185]. CCL11, a multifunctional chemokine, has a dual role in cancer, demonstrating potential in immune therapy [186]. CXCL5 has been also identified as a key chemokine driving the infiltration of mature pro-tumor neutrophils into lung cancer tissues [187]. Nevertheless, precisely blocking these chemokines without affecting systemic immune function remains a challenge. Additionally, sialic acid-modified nanoplatforms can interfere with the physical or molecular interactions between neutrophils and the tumor microenvironment, blocking their infiltration and enhancing the efficacy of checkpoint blockade therapies [188]. Another strategy involves the direct clearance of pro-tumor myeloid cells, such as PMN-MDSCs, using near-infrared immunotherapy with Ly6G antibodies to selectively eliminate neutrophils in the tumor bed, significantly inhibiting tumor growth and enhancing host immune responses [189,190]. Further studies have shown that pathogens in the tumor microenvironment, such as Fusobacterium nucleatum, recruit TANs by activating the IL-17/NF-κB/RelB pathway, promoting gastric cancer progression, suggesting that targeting the microbiome may indirectly modulate neutrophil behavior [191]. These strategies provide diverse pathways for precisely intervening in neutrophil recruitment.

### 6.2. Regulation of Neutrophil-Derived Cytokine Release

Neutrophils, as critical effector cells in inflammation and immune regulation, can synthesize and secrete various cytokines, including pro-inflammatory cytokines (TNF-α, IL-1β), chemokines (IL-8, IP-10, MIP-1α), and angiogenic factors (VEGF). These factors not only promote inflammation and immune responses within the tumor microenvironment but also accelerate tumor progression [192]. Therefore, inhibiting the secretion of these cytokines by neutrophils is considered a potential strategy for immunotherapy. TGF-β1 activates the SMAD3 and ERK1/2 signaling pathways, inducing neutrophils in the tumor microenvironment to express tumor-promoting factors, such as OSM and VEGFA mRNA, thereby converting neutrophils into a pro-tumor phenotype [193]. Targeting the TGF-β1 signaling pathway can inhibit the pro-tumor functions of neutrophils, offering a new therapeutic target for precise modulation of the tumor microenvironment.

### 6.3. Bispecific Antibodies Enhance Neutrophil-Mediated Antitumor Activity

The novel bispecific antibody (TrisomAb) effectively recruits neutrophils as effector cells, enhancing their cytotoxicity against tumor cells. Studies have shown that, in colorectal cancer patients, when neutrophils are able to efficiently eliminate tumor cells are exposed to anti-EGFR TrisomAb [194].

### 6.4. Innovative Carriers for Tumor-Targeted Nanodrug Delivery

Neutrophils, as core effector cells of innate immunity, have become ideal carriers for targeted nanodrug delivery in brain tumors due to their rapid inflammatory response, strong chemotaxis, and ability to cross the blood-brain barrier (BBB) [195,196,197]. These cells actively participate in inflammation by forming NETs and releasing cytokines, providing a new mechanism for precise drug delivery [195] (Table 3). In conditions of neuroinflammation or pathological states, the integrity of the BBB is compromised, which facilitates neutrophil migration into the central nervous system (CNS) [198,199]. Acting as “Trojan horses,” neutrophils can carry nanodrugs and use their pro-inflammatory properties to deliver anti-cancer drugs precisely to residual glioma cells after surgery, significantly inhibiting tumor recurrence and extending mouse survival [197]. Biomimetic nanocarriers, designed to mimic the surface characteristics of neutrophils, possess immune evasion and targeting capabilities, effectively overcoming the limitations of the BBB and the tumor microenvironment, thereby enhancing drug accumulation and therapeutic efficacy. In a breast cancer model, these carriers demonstrated potential in modulating the microenvironment and inhibiting metastasis, offering new insights into brain tumor treatment [200,201]. Additionally, neutrophil-derived exosomes (NEs-Exos/DOX) leverage their strong inflammatory chemotaxis and BBB penetration ability to successfully deliver doxorubicin to gliomas, significantly inhibiting tumor growth, extending survival, and reducing systemic toxicity [202]. In a breast cancer model, local inflammation induced the recruitment of neutrophils and the release of NETs, further enhancing anti-tumor effects. At the same time, neutrophil infusion helped alleviate systemic inflammation, ensuring clinical safety [203] (Table 3). These studies collectively highlight the unique advantages of neutrophils and their derived carriers in overcoming the BBB barrier and enhancing targeted brain tumor therapy, providing innovative strategies for precision treatment and suggesting substantial clinical potential.

### 6.5. Neutrophil-Based Combination Therapy

#### 6.5.1. Combination with Immune Checkpoint Inhibitors

In a glioma mouse model, neutrophil depletion combined with anti-PD-1 antibody treatment significantly inhibited glioma growth and extended mouse survival [204]. In a non-small cell lung cancer (NSCLC) mouse model, CXCL5 gene knockout completely prevented neutrophil accumulation in the lung tissue, which in turn promoted the expansion and cytolytic function of tumor-specific CD8+ T cells [187]. Additionally, studies have shown that the synergistic action of T cells and neutrophils can enhance the efficacy of immunotherapy [205].

#### 6.5.2. Combination with Chemotherapy

In pancreatic cancer patients, combining anti-Ly6G therapy to deplete neutrophils with gemcitabine/paclitaxel chemotherapy not only significantly reduced tumor burden and metastatic growth but also enhanced the functionality of tumor-infiltrating CD8+ T cells, effectively suppressed the polarization of CAF, and inhibited chemotherapy resistance via the IL-6/STAT-3 signaling pathway [206].

#### 6.5.3. Other Combination Strategies with Neutrophil Modulators

The role of neutrophils in cancer immunotherapy has been significantly enhanced through various pharmacological and biological interventions. Pharmacological interventions, such as angiotensin-converting enzyme inhibitors (ACEIs) and angiotensin II type 1 receptor (AGTR1) antagonists, can drive neutrophil polarization toward the antitumor N1 phenotype, thereby enhancing their cytotoxic potential [140]. Clinically, increased intratumoral infiltration of N1 neutrophils has been correlated with improved patient outcomes. For example, suppression of the TGF-β signaling pathway has been shown to reprogram neutrophils into the N1 phenotype, leading to effective tumor suppression [141,142,143,144,145,146,147,148]. Computational modeling has further refined the regulation of N1/N2 neutrophil balance by TGF-β inhibitors and IFN-β, favoring N1 recruitment and significantly impeding tumor progression [207]. Moreover, melatonin has been found to enhance antitumor immunity through modulation of tumor-associated neutrophil infiltration and NETosis in pancreatic ductal adenocarcinoma [208]. Probiotics have also been implicated in exerting therapeutic benefits through the inhibition of neutrophil-mediated cancer metastasis [209].

### 6.6. Neutrophil-Lymphocyte Ratio (NLR): A Critical Biomarker for Cancer Prognosis Evaluation

Although clinical treatments targeting neutrophils are still in the exploratory stage, several studies in preclinical models and early clinical observations suggest that neutrophil-targeted therapies may improve patient prognosis. Research has shown that an increase in peripheral blood neutrophil count or NLR is an independent predictor of poor prognosis in various cancers. For example, in colorectal cancer patients, elevated neutrophil count is associated with a shorter overall survival (OS) [210]. In glioma patients, the presence of CTCs post-surgery is closely linked to neutrophil-mediated inflammatory immune environments and correlates with poor prognosis [211]. In NSCLC patients, persistent neutrophil elevation and a high NLR (≥5) are associated with negative prognostic value [212]. Additionally, in gastric cancer patients receiving PD-1 antibody treatment, high NLR is also linked to adverse outcomes [180]. These findings highlight the critical role of neutrophils in cancer progression and suggest their potential as therapeutic targets.

In pancreatic cancer patients, dynamic changes in NLR before and during chemotherapy are closely associated with pathological response, disease-free survival (DFS), and OS [206]. Some studies have also proposed that a TICO regimen, consisting of tadalafil, isotretinoin, colchicine, and Omega-3 fatty acids, may inhibit cancer growth and metastasis by reducing high NLR [213]. Although clinical data for this regimen still require validation through large-scale trials, its theoretical foundation and the safety of the drugs involved provide promising possibilities for future clinical applications.

Neutrophils and related indicators, such as NLR, not only serve as prognostic biomarkers but may also guide personalized treatment. Monitoring neutrophils in the tumor microenvironment is crucial for tumor treatment and prognosis assessment. Advances in molecular imaging technologies now enable non-invasive visualization of neutrophil biological behavior, offering strong support for personalized cancer diagnostics and therapeutic strategies [214].

**Table 3 biomedicines-13-01473-t003:** The role of NETs in different cancer types.

Tumor Type	Mechanisms	References
Breast cancer	Promote tumor cell proliferation, intravascular infiltration, and distant metastasis, PMN formation, and awakening of dormant cancer cells.	[215,216,217,218,219]
Neuroblastoma	Play a key role in tumor cell proliferation, metastasis, and immune escape.	[220]
Oropharyngeal squamous cell carcinoma	In the initial stages of tumor formation, it may suppress tumorigenesis by clearing bacteria; after the tumor is established, it promotes tumor progression by releasing cytokines and chemokines.	[221]
Gastric cancer	Promote tumor progression (e.g., IL-8-induced NETs); facilitate EMT and metastasis through the PAI-1/TGF-β axis.	[222,223,224]
Colorectal cancer	Promote tumor progression and liver metastasis.	[225,226,227,228,229]
Esophageal squamous cell carcinoma	Promote tumor cell proliferation, migration, invasion, and angiogenesis; CCDC25 is associated with poor prognosis.	[230,231]
Osteosarcoma	High levels of NETs formation in diagnostic biopsies are associated with poor response to neoadjuvant chemotherapy and worse overall survival; NETs levels are higher in metastatic sites than in primary lesions.	[232,233]
Ovarian cancer	Promote the formation of the pre-metastatic niche in the peritoneum.	[234]
Pancreatic adenocarcinoma	Melatonin has been found to enhance anti-tumor immunity by regulating TANs infiltration and NETosis.	[208]

## 7. Conclusions and Future Perspectives

In summary, the functional heterogeneity and plasticity of neutrophils play a pivotal role in tumor initiation, progression, and metastasis. These cells exhibit a dual role: either promoting inflammatory responses and tumor metastasis, or acquiring anti-tumorigenic phenotypes under specific conditions. This functional duality positions neutrophils as critical regulators within the tumor microenvironment, highlighting their potential as targets for innovative anti-cancer therapeutic strategies.

However, current research faces several limitations and challenges: 1. Unclear Mechanisms: the intricate molecular mechanisms underlying the interactions among neutrophils, tumor cells, and other immune components, along with their evolving roles during cancer progression, are not yet fully elucidated. 2. High Variability: neutrophil behavior displays significant heterogeneity across different cancer types, individual patients, and tumor stages. This variability complicates mechanistic studies and limits the development of broadly applicable therapeutic strategies. 3. Technical Limitations: current methodologies, including single-cell sequencing and advanced imaging technologies, are often expensive, technically challenging, and may lack standardization. Accurately characterizing neutrophil functional states and identifying tumor-specific subpopulations remains a major technical barrier.

To address these limitations, future research should focus on the following areas:1. In-depth Mechanistic Studies: Utilizing cutting-edge approaches such as gene editing (e.g., CRISPR-Cas9) and multi-omics profiling to dissect the precise and context-dependent interactions between neutrophils, tumor cells, and immune cells. These efforts aim to uncover both shared and cancer-specific molecular mechanisms. 2. Clinical Translation: Building upon deeper mechanistic understanding, integrating artificial intelligence and big data analytics to identify novel diagnostic and therapeutic targets. This will facilitate the development of personalized precision medicine strategies for early cancer detection, diagnosis, and monitoring of treatment responses. 3. Technological Advancements: Advancing the development of cost-effective, user-friendly tools—such as high-throughput sequencing platforms and high-resolution imaging systems—to improve accessibility of affordable diagnostics and alleviate the clinical burden of cancer.

## Figures and Tables

**Figure 1 biomedicines-13-01473-f001:**
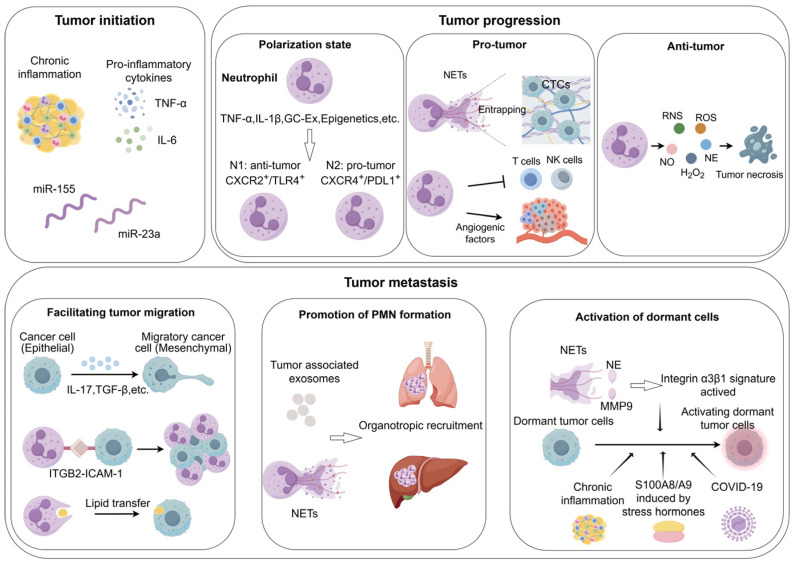
The function of neutrophils in tumors. Neutrophils have varied roles across tumor stages. During tumor initiation, they can foster tumorigenesis via chronic inflammation, pro-inflammatory cytokines, and non-coding RNAs. As the tumor progresses, they can either support or hinder tumor growth via mechanisms including N1/N2 polarization, NETs, angiogenesis modulation, and the release of cell-killing substances. In the metastatic phase, neutrophils promote the dissemination of tumor cells to distant sites by facilitating tumor migration, promotion of PMN formation, and activation dormant tumor cells.

**Figure 2 biomedicines-13-01473-f002:**
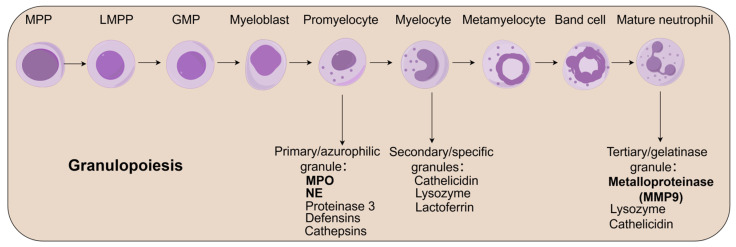
Neutrophil granulopoiesis and granule changes. Neutrophil development begins with lymphoid-primed multipotent progenitors (LMPP). These cells, derived from multipotent progenitors (MPP), first differentiate into granulocyte-monocyte progenitors (GMP) during development. The GMP gives rise to the myeloblast, the initial recognizable myeloid form. Myeloblasts mature into promyelocytes, which acquire primary/azurophilic granules, then into myelocytes, which have secondary/specific granules and an eccentric nucleus. These develop into metamyelocytes and subsequently into band cells, characterized by a band-shaped, non-segmented nucleus. The final maturation step sees the band cell’s nucleus segmenting to form the phagocytic mature neutrophil, which acquires tertiary/gelatinase granules. Neutrophil granules include three main types: primary/azurophilic, containing MPO, NE, proteinase 3, defensins, and cathepsins; secondary/specific, which contain cathelicidin, lysozyme, and lactoferrin; and tertiary/gelatinase, containing lysozyme and cathelicidin.

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
