# Peer review of "Neutrophil Spatiotemporal Regulatory Networks: Dual Roles in Tumor Growth Regulation and Metastasis"

_biomedicines, 2025, doi:10.3390/biomedicines13061473_

Round 1

Reviewer 1 Report

Comments and Suggestions for Authors

The authors pointed out (line 111) in this very good review that neutrophil lifespan can be extended to 24-48 hours before spontaneous apoptosis, with certain microenvironments further delaying apoptosis. However, they did not discuss this phenomenon in relation to the tumor microenvironment. Are there any data on neutrophil lifespan in cancer?

Please remove (or modify) a legend for Table 1 (lines 443-447).

Please check the list of links. Some references are truncated (for example, 29, 38, 44, 96, 185, etc.).

Please use the same format for references (eg reference 162).

Author Response

Comment 1: The authors pointed out (line 111) in this very good review that neutrophil lifespan can be extended to 24-48 hours before spontaneous apoptosis, with certain microenvironments further delaying apoptosis. However, they did not discuss this phenomenon in relation to the tumor microenvironment. Are there any data on neutrophil lifespan in cancer?

Response 1:Thank you very much. In response to your question, I conducted extensive literature review, and tumors can indeed affect neutrophil lifespan. The specific description has been added in lines 111-128 of the article, marked with underlines.

line 111-128: The TME markedly extends neutrophil lifespan through diverse mechanisms, promoting tumor progression. Cytokines like granulocyte colony-stimulating factor (G-CSF) activate the NF-κB pathway, inducing CD54 and B7-H6 expression on neutrophils, suppressing apoptosis, and prolonging survival; tumor tissue culture supernatant exerts similar effects in a time- and dose-dependent manner[55]. Specialized pro-resolving mediators (SPMs) induce programmed neutrophil death, enhance macrophage clearance of apoptotic cells, and inhibit infiltration and degranulation, aiding inflammation resolution and tissue homeostasis[56]. Although focused on kidney disease, this study’s insights into apoptosis regulation may have broader relevance, offering potential targets for modulating neutrophil lifespan in the TME. Pathological conditions in the TME, such as hypoxia and nutrient deprivation, alter neutrophil metabolism, disrupting spontaneous apoptosis and extending survival[57]. Tumor-derived small extracellular vesicles (sEVs) carrying serine protease inhibitors (serpins) reprogram neutrophils toward a pro-tumorigenic phenotype, further prolonging their lifespan[58]. Collectively, the TME systematically inhibits neutrophil apoptosis via cytokines, metabolic reprogramming, and sEVs, extending lifespan, sustaining a pro-tumorigenic inflammatory microenvironment, and driving tumor progression.”

Comment 2:Please remove (or modify) a legend for Table 1 (lines 443-447).

 Response 2:Thank you for your comments. Regarding your inquiry, we have removed the legend for Table 1 as you noted.

Comment 3:Please check the list of links. Some references are truncated (for example, 29, 38, 44, 96, 185, etc.).

 Response 3:Thank you for pointing out the errors. I have corrected the format of these references. Due to changes in other content, the order of these references has changed, and I have marked them with underlines.

“27. Townsend, M.; Galbraith, M.A.; Ellber, R. Selective Reduction of Human Tumor Cell Populations by Human Granulocytesin Vitro. Cancer Res. 1978, 38, 4534-4539, PMID: 569013.

28. Gerrard, T.L.; Kaplan, A.M. Human Neutrophil-Mediated Cytotoxicity to Tumor Cells. J Natl Cancer Inst. 1981, 66, 483-488, PMID: 6937705.

38. Wislez, M.; Rabbe, N.; Marchal, J.; Milleron, B.; Crestani, B.; Mayaud, C.; Antoine, M.; Soler, P.; Cadranel, J. Hepatocyte Growth Factor Production by Neutrophils Infiltrating Bronchioloalveolar Subtype Pulmonary Adenocarcinoma: Role in Tumor Progression and Death. Cancer Res. 2003, 63, 1405-1412, PMID: 12649206.

44. Fiedler, K.; Brunner, C. The Role of Transcription Factors in the Guidance of Granulopoiesis. Am J Blood Res. 2012, 2, e57-65, PMID: 22432088.

100. Apte, R.N.; Segal, S.; Dinarello, C.A.; Song, X.; Krelin, Y.; Dvorkin, T. Bearing Tumors of IL-1β-Secreting Cells Mediate Suppression of T Cells in Mice CD11b+/Gr-1+ Immature Myeloid Cells. J Immunol. 2005, 175, 8200-8208, doi: 10.4049/jimmunol.175.12.8200.

161. Mittal, V. Epithelial Mesenchymal Transition in Tumor Metastasis. Annu Rev Pathol. 2018, 13, 395-412, doi: 10.1146/annurev-pathol-020117-043854.

192. Nieto, P.; Elosua-Bayes, M.; Trincado, J.L.; Marchese, D.; Massoni-Badosa, R.; Salvany, M.; Henriques, A.; Nieto, J.; Aguilar-Fernández, S.; Mereu, E.; et al. A Single-Cell Tumor Immune Atlas for Precision Oncology. Genome Res 2021, 10,1913-1926, doi: 10.1101/gr.273300.120”

Comment 4:Please use the same format for references (eg reference 162).

 Response4 :Thank you for your helpful comment regarding.I have updated the format of all references to be consistent with that of reference 162.

Reviewer 2 Report

Comments and Suggestions for Authors

In the manuscript, the authors can add the recent advances in therapeutic approaches in brain cancer or others such as using neutrophils as a carrier of therapeutic nano-particles in order to overcome the blood-brain barrier (BBB). 

Author Response

Comment :In the manuscript, the authors can add the recent advances in therapeutic approaches in brain cancer or others such as using neutrophils as a carrier of therapeutic nano-particles in order to overcome the blood-brain barrier (BBB). 

Response: Thank you for your helpful suggestions. To address your inquiry more thoroughly, I undertook an in-depth literature review focusing on the application of neutrophils as nanocarriers in delivering tumor therapies across the blood-brain barrier, which I have since synthesized into a coherent summary. This content has been incorporated into the section on Targeting Neutrophils in Cancer Therapy: Strategies and Approaches (lines 542–567).

lines 540–565: 8.4 Innovative carriers for tumor-targeted nanodrug delivery Neutrophils, as core effector cells of innate immunity, have become ideal carriers for targeted nanodrug delivery in brain tumors due to their rapid inflammatory response, strong chemotaxis, and ability to cross the blood-brain barrier (BBB) [205–207]. These cells actively participate in inflammation by forming NETs and releasing cytokines, providing a new mechanism for precise drug delivery [205]. In conditions of neuroinflammation or pathological states, the integrity of the BBB is compromised, which facilitates neutrophil migration into the central nervous system (CNS) [208,209]. Acting as "Trojan horses," neutrophils can carry nanodrugs and use their pro-inflammatory properties to deliver anti-cancer drugs precisely to residual glioma cells after surgery, significantly inhibiting tumor recurrence and extending mouse survival [210]. Biomimetic nanocarriers, designed to mimic the surface characteristics of neutrophils, possess immune evasion and targeting capabilities, effectively overcoming the limitations of the BBB and the tumor microenvironment, thereby enhancing drug accumulation and therapeutic efficacy. In a breast cancer model, these carriers demonstrated potential in modulating the microenvironment and inhibiting metastasis, offering new insights into brain tumor treatment [211,212]. Additionally, neutrophil-derived exosomes (NEs-Exos/DOX) leverage their strong inflammatory chemotaxis and BBB penetration ability to successfully deliver doxorubicin to gliomas, significantly inhibiting tumor growth, extending survival, and reducing systemic toxicity [213]. In a breast cancer model, local inflammation induced the recruitment of neutrophils and the release of NETs, further enhancing anti-tumor effects. At the same time, neutrophil infusion helped alleviate systemic inflammation, ensuring clinical safety [214]. These studies collectively highlight the unique advantages of neutrophils and their derived carriers in overcoming the BBB barrier and enhancing targeted brain tumor therapy, providing innovative strategies for precision treatment and suggesting substantial clinical potential.”

Reviewer 3 Report

Comments and Suggestions for Authors

The manuscript "Neutrophil Spatiotemporal Regulatory Networks: Dual Roles in Tumor Progression and Metastasis" by Li and colleagues provides a comprehensive and up-to-date synthesis of the dual roles of neutrophils in tumor biology, spanning tumor initiation, progression, and metastasis. The review thoroughly explores neutrophil biology, from development (granulopoiesis) to their dichotomous roles in cancer (anti-tumor N1 vs. pro-tumor N2 subsets).The authors highlight the therapeutic potential of targeting neutrophils in cancer treatment while acknowledging unresolved challenges. Here are some comments listed below:

1.The title implies a focus on “spatiotemporal regulatory networks,” but no network modeling or spatial transcriptomic/spatial proteomic data are presented or reviewed.

2. This is an exciting and relatively area, but it's not deeply developed in the review. What are the experimental models or clinical datasets supporting this? Any mention of single-cell or spatial analyses that show reactivation patterns?

Author Response

Comment 1:The title implies a focus on “spatiotemporal regulatory networks,” but no network modeling or spatial transcriptomic/spatial proteomic data are presented or reviewed.

Response 1: Thank you for your helpful comment regarding. To advance our understanding of tumor-associated neutrophil (TAN) heterogeneity in cancer, I performed an in-depth review of the literature focusing on single-cell sequencing and spatial omics studies, synthesizing the key findings. The information and conclusions have been compiled into a dedicated section, titled "7. Single-Cell and Spatial Omics Uncover Neutrophil Heterogeneity and Therapeutic Targets," detailed in lines 453–495.

"lines 453–494: 7.Single-Cell and Spatial Omics Uncover Neutrophil Heterogeneity and Therapeutic Targets

Single-cell and spatial omics have elucidated the precise localization and functional roles of neutrophils within the TME across various cancer types, highlighting their heterogeneity and interactions with other cellular components. In Prostate cancer, integrated scRNA sequencing and high-resolution spatial transcriptomics revealed an immunosuppressive TME driven by myeloid cells, including neutrophils, and exhausted T cells, contributing to tumor recurrence and metastasis [184]. In Non-small cell lung cancer (NSCLC), single-cell and spatial transcriptomic analyses underscored the pervasive presence of myeloid cells, including neutrophils, and their pivotal role in disease progression, providing a high-resolution molecular map of their contributions to the TME [185]. For Pancreatic ductal adenocarcinoma (PDAC), multi-omics analyses have elucidated the heterogeneity of neutrophils in the PDAC tumor microenvironment at single-cell resolution, identifying four distinct neutrophil subsets and revealing their dynamic functions, metabolic reprogramming, and regulatory mechanisms [186]. Additionally, single-cell and spatial transcriptomic studies have characterized multicellular dynamics associated with neoadjuvant therapy in PDAC, providing a framework for understanding neutrophil contributions to the spatial TME, despite not directly addressing them [187]. In Ovarian cancer, a microfluidic-based TIME-on-Chip model demonstrated that NETs promote collective tumor invasion, offering an in vitro platform to study neutrophil spatial functions [188]. In glioblastoma: single-cell RNA sequencing highlighted the emergence of pro-tumor myeloid-derived suppressor cells (MDSCs), including neutrophils, during disease progression [189]. In Gastric cancer, single-cell RNA sequencing post-PD-1 therapy identified increased circulating neutrophils, with the NE-1 subset exhibiting an activated phenotype (high MMP9, S100A8, S100A9, PORK2, TGF-β1 expression) and interacting with malignant epithelial cells and M2 macrophages via chemokine pathways to promote tumor progression [190]. Pan-cancer analyses across 17 cancer types and 225 samples revealed transcriptional heterogeneity among tumor-associated neutrophils, forming 10 distinct states; HLA-DR+CD74+ antigen-presenting neutrophils correlated with improved survival, while VEGFA+SPP1+ angiogenic neutrophils were associated with poorer prognosis. Interventions like leucine diets or delivery of antigen-presenting neutrophils enhanced anti-PD-1 immunotherapy in mouse models and human tumor fragment models, suggesting novel neutrophil-based therapeutic strategies [191]. Additionally, a single-cell tumor immune atlas from 217 patients across 13 cancers, integrated with the SPOTlight tool, enabled in situ immune population mapping, supporting patient stratification based on immune composition [192].

Spatial omics, integrated with single-cell omics, will advance our understanding of neutrophil heterogeneity, spatial distribution, and their complex interactions with tumor progression and therapeutic responses within the tumor microenvironment. These high-resolution molecular profiles, coupled with in vitro models simulating three-dimensional tumor microenvironments, will provide a robust foundation for developing precise neutrophil-targeted anticancer therapies."

Comment 2: This is an exciting and relatively area, but it's not deeply developed in the review. What are the experimental models or clinical datasets supporting this? Any mention of single-cell or spatial analyses that show reactivation patterns?

Response 2: Thank you for your helpful comment regarding.In response to your inquiry, I will systematically provide detailed explanations for each of the references in Chapter 7.

Reference 184:This article, through integrated scRNA sequencing and high-resolution spatial transcriptomics, revealed an immunosuppressive tumor microenvironment (TME) driven by myeloid cells, including neutrophils, and exhausted T cells, contributing to tumor recurrence and metastasis. The samples in this study were derived from patients with clinically localized prostate cancer who underwent minimally invasive transabdominal radical prostatectomy. The raw data uploaded by the authors is available under the data accession number GSE181294.

Reference 185:In this study, the authors analyzed clinical samples comprising tumor tissues and matched adjacent non-tumor tissues collected from the lungs of 25 patients after marginal resection, along with healthy lung tissues obtained from two postmortem donors, using single-cell and spatial transcriptomic approaches. These analyses demonstrated that myeloid cells, including neutrophils, are prevalent within the tumor microenvironment of non-small cell lung cancer and play critical roles in disease progression. Moreover, this multi-omics dataset offers a high-resolution molecular atlas of myeloid cells, significantly advancing our understanding of their functional roles within the tumor microenvironment. The raw sequencing data have been deposited in the public repository under accession numbers E-MTAB-13526 and E-MTAB-13530.

Reference 186:The study employs tumor tissues and peripheral blood samples from PDAC patients to isolate CD66-positive neutrophils via fluorescence-activated cell sorting (FACS) for downstream analysis. Through integrated multi-omics analyses—including scRNA-seq, spatially resolved transcriptomics, bulk transcriptomics, proteomics, and metabolomics—this study provides the first comprehensive single-cell-level characterization of TAN dynamics, metabolic reprogramming, and molecular regulatory mechanisms within the PDAC tumor microenvironment. The raw sequencing data have been deposited in the National Omics Data Encyclopedia (NODE) under accession number OEP003254.

Reference 188:In this study, the authors employed a microfluidic chip to recapitulate the tumor immune microenvironment ( TIME-on-Chip). The researchers successfully reconstructed the three-dimensional (3D) dynamic interactions between neutrophils and ovarian cancer cells (OVCAR-3), revealing that neutrophil extracellular traps (NETs) facilitate collective tumor invasion, potentially highlighting a pro-invasive role of NETs in ovarian cancer progression. This in vitro model provides new insights into the spatial dynamics of neutrophil involvement in tumor invasion. It is important to note that this article does not include original experimental raw data.

Reference 189:In this investigation, the researchers conducted single-cell RNA sequencing (scRNA-seq) using clinical specimens from patients diagnosed with glioblastoma (GBM), as well as a genetically engineered murine model characterized by conditional overexpression of human EGFR combined with Cdkn2a and Pten deletion. The study elucidates dynamic changes in the tumor immune landscape during glioblastoma progression, with particular emphasis on the infiltration and functional role of pro-tumorigenic myeloid-derived suppressor cells (MDSCs). Raw sequencing data have been made publicly available through the Gene Expression Omnibus (GEO) repository under the following accession codes: GSE195848, GSE196174, GSE196175, and GSE195813.

Reference 190:The research team conducted single-cell RNA sequencing on pretreatment tumor biopsies from advanced gastric cancer patients receiving systemic immunotherapy. During anti-PD-1 immunotherapy, circulating neutrophil counts significantly expanded in peripheral blood mononuclear cell compartments. The study identified a distinct neutrophil subset (Neutrophil Cluster 1, NE-1) characterized by activation markers and functional interactions with tumor-associated epithelial cells and M2-polarized macrophages via chemokine-mediated signaling mechanisms. The findings suggest that this neutrophil-tumor-macrophage crosstalk contributes to disease progression. Data are publicly available in the NCBI BioProject database (Accession No. PRJNA975683).

Reference 191:This study provides a comprehensive characterization of the complexity and heterogeneity of tumor-associated neutrophils through single-cell RNA-seq and spatial transcriptomics of 225 samples across 17 cancer types. Neutrophils display marked transcriptional heterogeneity across various cancer types. Notably, a distinct subset of HLA-DR+ CD74+ neutrophils with antigen-presenting function was identified, which correlates with better clinical outcomes in patients. The raw sequencing data have been deposited in the China National Center for Bioinformation (accession number: PRJCA020880) and are publicly available.

Reference 192:This study integrates data from over 500,000 cells representing 217 patients across 13 cancer types sourced from public databases. We developed the SPOTlight algorithm, integrating single-cell RNA sequencing (scRNA-seq) and spatial transcriptomic data, to enable in situ localization of immune cell subsets. The processed datasets have been deposited in the NCBI Gene Expression Omnibus (GEO) under accession number GSE158803.

Reviewer 4 Report

Comments and Suggestions for Authors

This review focuses on the role of neutrophils in tumorigenesis and antitumor defence. The data presented in this review may interest a wide range of readers, as they address the underlying mechanisms of tumorigenesis and outline possible new immunotherapeutic approaches to the antitumor treatment. Nevertheless, the MS has notable flaws and warrants improvement.

My comments and suggestions.

Major:

1. The MS is poorly structured. There are many returns to what has already been said. The principle of dividing the MS into sections and subsections is unclear. Sections of the MS should consistently detail neutrophils’ involvement in carcinogenesis, tumor progression, and metastasis. A dedicated section might focus on neutrophil antitumor properties and how to boost them.

2. Table 1 is overloaded with information. It would be useful for a better understanding to divide this table into several tables, which are organized into appropriate sections.

3. Chapter 4 “Neutrophils serve as a major nexus bridging inflammation...” addresses the role of chronic inflammation in carcinogenesis. However, nothing is said here about the importance of reactive oxygen species (ROS) in this process. It is well known that neutrophil-derived ROS provide carcinogenic mutagenesis in foci of chronic inflammation (Roessner A, Kuester D, Malfertheiner P, Schneider-Stock R. Oxidative stress in ulcerative colitis-associated carcinogenesis. doi: 10.1016/j.prp.2008.04.011. Epub 2008 Jun 20. PMID: 18571874;Wu S, Chen Y, Chen Z, Wei F, Zhou Q, Li P, Gu Q. Reactive oxygen species and gastric carcinogenesis: The complex interaction between Helicobacter pylori and host. Helicobacter. 2023 Dec;28(6):e13024. doi: 10.1111/hel.13024. Epub 2023 Oct 5. PMID: 37798959).

4. The conclusion section is not really a conclusion, but is essentially a brief restatement of what was previously said. It lacks generalizations, conclusions, and future perspectives.

5. Since neutrophils are the dominant population of immune cells, they are the ones that may have the greatest impact on tumor growth. However, very little attention has been paid in the MS to the immunotherapeutic possibilities of modulating the functional activity of neutrophils.

Minor:

1. Figures 3 and 4 are overloaded with information. The captions to these figures are not informative and are not clearly delineated from the main text of the MS. It would be useful to divide this figure into several figures and place these figures in the appropriate sections.

2. It was interesting and useful to discuss the interaction of neutrophils with other granulocyte subpopulations (eosinophils and basophils) in this article. Unfortunately, the role of this interaction in both tomologenesis and antitumor defense is still grossly underreported.

Author Response

Comment 1:Major1. The MS is poorly structured. There are many returns to what has already been said. The principle of dividing the MS into sections and subsections is unclear. Sections of the MS should consistently detail neutrophils’ involvement in carcinogenesis, tumor progression, and metastasis. A dedicated section might focus on neutrophil antitumor properties and how to boost them.

Response 1: Thank you for your comments.  We unanimously agree that this suggestion is highly professional.We have restructured the manuscript to enhance clarity and coherence, streamlining overlapping or redundant content. The revised sections have been reorganized and retitled to reflect the distinct roles of neutrophils in tumorigenesis, progression, and metastasis. Furthermore, we have included a dedicated discussion on the anti-tumor functions of neutrophils.The modified sections are marked with underlines in the manuscript.

line 210: 4. Neutrophils as Key Mediators of Inflammation-to-Carcinogenesis 

line 257: 5. The Dualistic Behavior of Neutrophils in the Tumor Progression.

line265-325: 5.1 Pro-Tumorigenic Roles of Neutrophils

5.1.1 Pro-angiogenesis: Neutrophils, particularly those polarized toward the N2 phenotype, are key contributors to tumor angiogenesis through the secretion of multiple pro-angiogenic factors, including FGF2, VEGF-A, ANGPT1, CXCL8, HGF, and MMP9 [118,119,121–132], which collectively facilitate tumor vascularization and metastatic spread. Studies have shown that metastasis-associated neutrophils exhibit markedly elevated FGF2 expression relative to their naïve counterparts, contributing to enhanced hepatic metastatic progression through increased vascular density and branching [121]. In addition, MMP9 secreted by these neutrophils further enhances tumor neoangiogenesis, thereby promoting the development of pancreatic and lung cancers [122]. Importantly, research has demonstrated that MMP9 released by bone marrow-derived neutrophils collaborates with other hematopoietic lineage cells, possibly by modulating the tumor microenvironment or extracellular matrix remodeling, to promote the initiation of squamous cell carcinoma [133].

5.1.2 NETs Formation: NETs promote tumor metastasis through both physical entrapment and molecular signaling mechanisms. Within their extracellular fibrous network, NETs capture circulating tumor cells (CTCs), shielding them from the cytotoxic effects of CD8+ T cells and NK cells while simultaneously facilitating metastatic adhesion [118,134,135]. Higher levels of circulating NETs have been shown to correlate with increased metastatic burden in advanced esophageal cancer, gastric cancer, and lung cancer [135]. Amyloid-beta, secreted by CAFs, promotes NETs formation and contributes to tumor progression [136]. At the mechanistic level, tumor cells utilize the CCDC25 receptor to recognize DNA associated with NETs, thereby activating the ILK-β-parvin signaling axis and enhancing the adhesion, motility, and proliferation of liver-metastatic cells [137]. This process accelerates the growth of liver metastases, providing direct evidence for the pro-metastatic role of NETs. Studies in mouse models further demonstrate that NETs infiltrating into peritumoral tissues can enhance metastatic potential [138,139]. Furthermore, inflammatory signals not directly linked to tumor initiation can also trigger neutrophil NETosis [140,141].

5.1.3 Immunosuppressive Microenvironment Remodeling: The suppression of T cell activity by neutrophils represents a critical mechanism underlying tumor immune escape, with distinct mechanisms observed in different tumor types: in colorectal cancer, N2-polarized neutrophils promote immune evasion by activating latent TGFβ via MMP9, thereby suppressing the function of tumor-infiltrating T cells [142]; whereas in ovarian cancer, these TANs mediate IL-8-dependent immunosuppression through the Jagged2 signaling pathway [143].

5.2 Anti-Tumorigenic Roles of Neutrophils

5.2.1 Oxidative stress-mediated tumor cell killing:  Neutrophils exert dual antitumor effects through a synergistic mechanism involving ROS and RNS. Mechanistic studies have demonstrated that H₂O₂ secreted by neutrophils activates the TRPM2 calcium channel on tumor cell membranes, leading to a rapid increase in intracellular Ca²⁺ levels. This elevation in Ca²⁺ subsequently initiates calpain-mediated activation of caspase-3, culminating in apoptotic cell death [144,145]. In addition, neutrophil-derived H₂O₂ has been shown to suppress pulmonary metastatic colonization [146]. Moreover, research by the Finisguerra group revealed that HGF interacts with the MET receptor on neutrophil surfaces, thereby activating the iNOS signaling axis and promoting NO production, which enhances tumor cell cytotoxicity [147]. Moreover, N1-polarized neutrophils potentiate anti-tumor immune responses and improve therapeutic outcomes via oxidative stress-mediated suppression of IL-17+ γδ T cells [148].

5.2.2 Protease-dependent selective cytotoxicity: NE demonstrates a distinct ability to target tumors and has been shown to selectively destroy cancer cells while inhibiting tumorigenesis [149]. This targeted cytotoxic effect works alongside oxidative mechanisms, thereby enhancing the overall antitumor response.

5.2.3 Pharmacological Enhancement of Tumor SuppressionPharmacologically, angiotensin-converting enzyme inhibitors (ACEIs) and angiotensin II type 1 receptor (AGTR1) antagonists promote neutrophil polarization toward the N1 phenotype, which enhances anti-tumor cytotoxicity [150]. Clinically, this effect is reflected in the association between intratumoral infiltration of N1 neutrophils and a favorable prognosis. For example, inhibition of TGF-β signaling can reprogram neutrophils into an N1-polarized state, thus effectively suppressing tumor growth [151–158].

lines 341:6.The Role of Neutrophils in Tumor Metastasis”

Comment 2: Major 2. Table 1 is overloaded with information. It would be useful for a better understanding to divide this table into several tables, which are organized into appropriate sections.

Response2 :Thank you for your helpful comment regarding. Following your recommendation, we have divided the original Table 1 into two distinct tables: Table 1 and Table 2. Table 1 now addresses the mechanisms by which neutrophils contribute to tumor progression, whereas Table 2 outlines their involvement in tumor metastasis. These tables have been moved to locations within the manuscript that better align with the content they support.

Comment 3: Major 3. Chapter 4 “Neutrophils serve as a major nexus bridging inflammation...” addresses the role of chronic inflammation in carcinogenesis. However, nothing is said here about the importance of reactive oxygen species (ROS) in this process. It is well known that neutrophil-derived ROS provide carcinogenic mutagenesis in foci of chronic inflammation (Roessner A, Kuester D, Malfertheiner P, Schneider-Stock R. Oxidative stress in ulcerative colitis-associated carcinogenesis. doi: 10.1016/j.prp.2008.04.011. Epub 2008 Jun 20. PMID: 18571874;Wu S, Chen Y, Chen Z, Wei F, Zhou Q, Li P, Gu Q. Reactive oxygen species and gastric carcinogenesis: The complex interaction between Helicobacter pylori and host. Helicobacter. 2023 Dec;28(6):e13024. doi: 10.1111/hel.13024. Epub 2023 Oct 5. PMID: 37798959).

Response 3 :Thank you for your helpful suggestions. We sincerely appreciate your valuable comments. We regret having previously overlooked the importance of reactive oxygen species (ROS) in promoting the transition from chronic inflammation to tumorigenesis. After thoroughly reviewing the two references you kindly provided, we have gained a deeper understanding and concluded that neutrophil-derived ROS significantly contribute to DNA damage and subsequent carcinogenic mutations within chronic inflammatory microenvironments. The corresponding revisions are clearly marked with underlines in the revised manuscript.

lines 234-251: Reactive oxygen species (ROS), particularly those derived from neutrophils, play a pivotal role in chronic inflammatory lesions by inducing carcinogenic mutations, with colitis serving as a representative example [116,117]. Helicobacter pylori infection and IBD induce extensive neutrophil infiltration and oxidative burst, generating ROS and RNS, including superoxide (O2˙), H2O2, hypochlorous acid (HOCl), hydroxyl radicals (˙OH), and peroxynitrite, which drive gastric and colorectal carcinogenesis. Produced primarily via NADPH oxidase and mitochondrial electron transport chain, ROS/RNS, while intended to eliminate pathogens, cause widespread host cell damage, including DNA lesions, lipid peroxidation, and protein oxidation. These insults provoke genomic instability, membrane dysfunction, and protein impairment, compromising cellular integrity. Additionally, ROS act as second messengers, activating NF-κB, AP-1, and matrix metalloproteinase-1 (MMP1), upregulating IL-8, VEGF, and iNOS, thereby promoting inflammation, proliferation, survival, angiogenesis, and tumor invasion/metastasis. Oxidative stress further induces epigenetic silencing of tumor suppressor genes (e.g., p53, Rb, hMLH1, p16, p14) via DNA methylation. Collectively, neutrophil-driven oxidative stress significantly contributes to H. pylori-associated gastric cancer and IBD-related colorectal cancer by damaging cellular components, modulating signaling pathways, and inducing genetic/epigenetic aberrations.”

Comment 4: Major 4.The conclusion section is not really a conclusion, but is essentially a brief restatement of what was previously said. It lacks generalizations, conclusions, and future perspectives.

Response 4 :Thank you for your helpful comment regarding.We sincerely appreciate your insightful suggestion. In response, we have substantially revised Chapter 9, titled "Conclusions and Future Perspectives," to enhance its academic rigor and clarity. Specifically, we have condensed the key findings in the Conclusions section while comprehensively analyzing the current challenges in tumor-associated neutrophil research within the Future Perspectives section. For each identified challenge, we have proposed practical strategies that could potentially advance this field. The revised portions are clearly indicated with underlines in the updated manuscript. We believe these modifications have substantially enhanced the overall quality and relevance of this chapter.

lines 621-680: 9.Conclusions and Future Perspectives

In conclusion, neutrophils, with their multifaceted biological functions and remarkable adaptability, profoundly influence every stage of tumor initiation, progression, and metastasis. They serve as key drivers of chronic inflammation and the formation of pro-metastatic microenvironments, while also possessing the capacity to transform into anti-tumor immune effector cells under specific conditions. This complex bidirectional role and functional heterogeneity establish neutrophils as central regulators within the tumor microenvironment, offering broad prospects for the development of innovative anti-tumor therapies targeting neutrophils.

However, current research faces several limitations and challenges: 1. Incomplete Mechanistic Understanding: Although neutrophils are known to play critical roles in tumor progression, immunosuppression, and metastasis, the underlying mechanisms remain incompletely elucidated. For instance, the precise interactions between neutrophils, tumor cells, and other immune cells—as well as their specific functions at different stages of tumor development—require further investigation. 2. Heterogeneity: Neutrophils exhibit significant heterogeneity across different tumor types, individual patients, and stages of tumor development. This variability complicates research and hinders the development of universal therapeutic strategies. 3. Limited Therapeutic Targets: Current neutrophil-targeted therapeutic approaches are limited, with efficacy varying by tumor type and patient. For example, therapies aimed at blocking neutrophil recruitment are effective in some cases but may be ineffective or cause adverse effects in others. 4. Insufficient Clinical Translation: Despite significant advances in basic research, translating these findings into clinical applications remains challenging. Key issues include accurately assessing neutrophil states and functions and leveraging them as therapeutic targets or biomarkers. 5. Technical Limitations: Existing research methods and technologies face constraints in dissecting neutrophil functions. While single-cell sequencing and spatial omics provide high-resolution data, they are costly and analytically complex. In vivo imaging techniques also lack precision in tracking dynamic neutrophil changes within the TME.

To address these shortcomings, future research on tumor-associated neutrophils will focus on the following directions to overcome current bottlenecks and advance clinical applications: 1. In-Depth Mechanistic Studies: Advanced technologies, such as single-cell sequencing, CRISPR-Cas9 gene editing, and high-throughput screening, will be employed to investigate the mechanistic roles of neutrophils in the TME. Key areas of focus will include their interaction networks with tumor cells and immune cells (e.g., T cells, macrophages), as well as their specific contributions to immune evasion, angiogenesis, and metastasis, aiming to uncover their critical functions in tumor development. 2. Heterogeneity Analysis: Through single-cell sequencing, spatial omics, and multi-omics integration, systematic studies will characterize neutrophil heterogeneity across tumor types, patient populations, and tumor stages. The objectives are to identify commonalities and differences among neutrophil subsets and provide a scientific basis for personalized and targeted therapies. 3. Novel Therapeutic Targets: Leveraging high-throughput screening, bioinformatics, and artificial intelligence-driven drug design, new therapeutic targets will be identified, and innovative therapies developed. These include antibodies targeting neutrophil surface receptors, cytokines or small-molecule drugs modulating neutrophil functions, and neutrophil-based immunotherapies—all aimed at enhancing treatment efficacy and specificity. 4. Clinical Translation: Efforts will focus on bridging basic research and clinical applications by developing neutrophil-related biomarkers for early tumor diagnosis (e.g., liquid biopsy techniques detecting neutrophil phenotypes or secretions), prognosis prediction (e.g., using neutrophil-associated gene expression profiles), and treatment monitoring (e.g., assessing changes in neutrophil states during therapy)—thereby facilitating the translation of research findings into clinical practice. 5. Technological Innovation: Advances in technology will be pursued to develop cost-effective and efficient tools. These include improved single-cell sequencing and spatial omics platforms to reduce costs and simplify data analysis, as well as high-resolution in vivo imaging techniques (e.g., nanoprobes or live imaging) to precisely track neutrophil dynamics in the TME, providing more accurate tools for studying neutrophil functions and behaviors.

These research directions aim to deepen the understanding of neutrophil roles in cancer, address current challenges, and pave the way for innovative diagnostic and therapeutic strategies in oncology.”

Comment 5:Major5. Since neutrophils are the dominant population of immune cells, they are the ones that may have the greatest impact on tumor growth. However, very little attention has been paid in the MS to the immunotherapeutic possibilities of modulating the functional activity of neutrophils.

Response 5: Thank you for your helpful suggestions. We appreciate your insightful suggestion and have accordingly added a new section titled Targeting Neutrophils in Cancer Therapy: Strategies and Approaches. This section offers an in-depth analysis of the diverse roles and functional mechanisms of neutrophils within the context of cancer immunotherapy. All revised sections have been clearly marked with underlines in the revised manuscript.

line 495-613: 8.Targeting Neutrophils in Cancer Therapy: Strategies and Approaches

TANs play a pivotal role in tumor progression. This section explores various strategies to target and inhibit TANs recruitment, as well as methods for monitoring and modulating neutrophil behavior to improve cancer treatment outcomes.

8.1 Strategies to inhibit TANs recruitment

The recruitment of TANs is a critical event in tumor progression, and targeting this process offers various strategies for cancer therapy. Chemokines and their receptors play an essential role in the migration of neutrophils to tumor sites. For example, CXCL2 promotes neutrophil recruitment and inflammatory responses via the CXCR2 axis, and inhibiting CXCR2 can alleviate myeloid inflammation and reverse treatment resistance in prostate cancer [193,194]. The ALX/FPR2 receptor regulates neutrophil infiltration based on ligands such as serum amyloid A, and targeting this receptor holds promise for controlling tumor-associated inflammation [195]. CCL11, a multifunctional chemokine, has a dual role in cancer, demonstrating potential in immune therapy [196]. CXCL5 has been also identified as a key chemokine driving the infiltration of mature pro-tumor neutrophils into lung cancer tissues [197]. Nevertheless, precisely blocking these chemokines without affecting systemic immune function remains a challenge. Additionally, sialic acid-modified nanoplatforms can interfere with the physical or molecular interactions between neutrophils and the tumor microenvironment, blocking their infiltration and enhancing the efficacy of checkpoint blockade therapies [198]. Another strategy involves the direct clearance of pro-tumor myeloid cells, such as PMN-MDSCs, using near-infrared immunotherapy with Ly6G antibodies to selectively eliminate neutrophils in the tumor bed, significantly inhibiting tumor growth and enhancing host immune responses [199,200]. Further studies have shown that pathogens in the tumor microenvironment, such as Fusobacterium nucleatum, recruit TANs by activating the IL-17/NF-κB/RelB pathway, promoting gastric cancer progression, suggesting that targeting the microbiome may indirectly modulate neutrophil behavior [201]. These strategies provide diverse pathways for precisely intervening in neutrophil recruitment.

8.2 Regulation of neutrophil-derived cytokine release

Neutrophils, as critical effector cells in inflammation and immune regulation, can synthesize and secrete various cytokines, including pro-inflammatory cytokines (TNF-α, IL-1β), chemokines (IL-8, IP-10, MIP-1α), and angiogenic factors (VEGF). These factors not only promote inflammation and immune responses within the tumor microenvironment but also accelerate tumor progression [202]. Therefore, inhibiting the secretion of these cytokines by neutrophils is considered a potential strategy for immunotherapy. TGF-β1 activates the SMAD3 and ERK1/2 signaling pathways, inducing neutrophils in the tumor microenvironment to express tumor-promoting factors, such as OSM and VEGFA mRNA, thereby converting neutrophils into a pro-tumor phenotype [203]. Targeting the TGF-β1 signaling pathway can inhibit the pro-tumor functions of neutrophils, offering a new therapeutic target for precise modulation of the tumor microenvironment.

8.3 Bispecific antibodies enhance neutrophil-mediated antitumor activity

The novel bispecific antibody (TrisomAb) effectively recruits neutrophils as effector cells, enhancing their cytotoxicity against tumor cells. Studies have shown that, in colorectal cancer patients, neutrophils, when exposed to anti-EGFR TrisomAb, are able to efficiently eliminate tumor cells [204].

8.4 Innovative carriers for tumor-targeted nanodrug delivery

Neutrophils, as core effector cells of innate immunity, have become ideal carriers for targeted nanodrug delivery in brain tumors due to their rapid inflammatory response, strong chemotaxis, and ability to cross the blood-brain barrier (BBB) [205–207]. These cells actively participate in inflammation by forming NETs and releasing cytokines, providing a new mechanism for precise drug delivery [205]. In conditions of neuroinflammation or pathological states, the integrity of the BBB is compromised, which facilitates neutrophil migration into the central nervous system (CNS) [208,209]. Acting as "Trojan horses," neutrophils can carry nanodrugs and use their pro-inflammatory properties to deliver anti-cancer drugs precisely to residual glioma cells after surgery, significantly inhibiting tumor recurrence and extending mouse survival [210]. Biomimetic nanocarriers, designed to mimic the surface characteristics of neutrophils, possess immune evasion and targeting capabilities, effectively overcoming the limitations of the BBB and the tumor microenvironment, thereby enhancing drug accumulation and therapeutic efficacy. In a breast cancer model, these carriers demonstrated potential in modulating the microenvironment and inhibiting metastasis, offering new insights into brain tumor treatment [211,212]. Additionally, neutrophil-derived exosomes (NEs-Exos/DOX) leverage their strong inflammatory chemotaxis and BBB penetration ability to successfully deliver doxorubicin to gliomas, significantly inhibiting tumor growth, extending survival, and reducing systemic toxicity [213]. In a breast cancer model, local inflammation induced the recruitment of neutrophils and the release of NETs, further enhancing anti-tumor effects. At the same time, neutrophil infusion helped alleviate systemic inflammation, ensuring clinical safety [214]. These studies collectively highlight the unique advantages of neutrophils and their derived carriers in overcoming the BBB barrier and enhancing targeted brain tumor therapy, providing innovative strategies for precision treatment and suggesting substantial clinical potential.

8.5 Neutrophil-Based Combination Therapy

8.5.1 Combination with immune checkpoint inhibitors: In a glioma mouse model, neutrophil depletion combined with anti-PD-1 antibody treatment significantly inhibited glioma growth and extended mouse survival [215]. In a non-small cell lung cancer (NSCLC) mouse model, CXCL5 gene knockout completely prevented neutrophil accumulation in the lung tissue, which in turn promoted the expansion and cytolytic function of tumor-specific CD8+ T cells [216]. Additionally, studies have shown that the synergistic action of T cells and neutrophils can enhance the efficacy of immunotherapy [217].

8.5.2 Combination with chemotherapy: In pancreatic cancer patients, combining anti-Ly6G therapy to deplete neutrophils with gemcitabine/paclitaxel chemotherapy not only significantly reduced tumor burden and metastatic growth but also enhanced the functionality of tumor-infiltrating CD8+ T cells, effectively suppressed the polarization of CAF, and inhibited chemotherapy resistance via the IL-6/STAT-3 signaling pathway[218].

8.5.3 Other combination strategies with nNeutrophil modulators: Pancreatic melatonin significantly enhanced anti-tumor immune responses by modulating tumor-associated neutrophil infiltration and NETosis in pancreatic adenocarcinoma [219]. Furthermore, mathematical models have been used to optimize the regulation of N1-N2 neutrophil dynamics with TGF-β inhibitors and IFN-β, further enhancing N1 recruitment and effectively inhibiting tumor progression [220]. Probiotics have also been proposed to potentially exert therapeutic effects by inhibiting neutrophil involvement in cancer metastasis [221].

8.6 Neutrophil-Lymphocyte Ratio (NLR): a critical biomarker for cancer prognosis evaluation

Although clinical treatments targeting neutrophils are still in the exploratory stage, several studies in preclinical models and early clinical observations suggest that neutrophil-targeted therapies may improve patient prognosis. Research has shown that an increase in peripheral blood neutrophil count or NLR is an independent predictor of poor prognosis in various cancers. For example, in colorectal cancer patients, elevated neutrophil count is associated with a shorter overall survival (OS) [222]. In glioma patients, the presence of CTCs post-surgery is closely linked to neutrophil-mediated inflammatory immune environments and correlates with poor prognosis [223]. In NSCLC patients, persistent neutrophil elevation and a high NLR (≥5) are associated with negative prognostic value [224]. Additionally, in gastric cancer patients receiving PD-1 antibody treatment, high NLR is also linked to adverse outcomes [225]. These findings highlight the critical role of neutrophils in cancer progression and suggest their potential as therapeutic targets.

In pancreatic cancer patients, dynamic changes in NLR before and during chemotherapy are closely associated with pathological response, disease-free survival (DFS), and OS [218]. Some studies have also proposed that a TICO regimen, consisting of tadalafil, isotretinoin, colchicine, and Omega-3 fatty acids, may inhibit cancer growth and metastasis by reducing high NLR [226]. Although clinical data for this regimen still require validation through large-scale trials, its theoretical foundation and the safety of the drugs involved provide promising possibilities for future clinical applications.

Neutrophils and related indicators, such as NLR, not only serve as prognostic biomarkers but may also guide personalized treatment. Monitoring neutrophils in the tumor microenvironment is crucial for tumor treatment and prognosis assessment. Advances in molecular imaging technologies now enable non-invasive visualization of neutrophil biological behavior, offering strong support for personalized cancer diagnostics and therapeutic strategies [227].“

Comment 6: Minor 1. Figures 3 and 4 are overloaded with information. The captions to these figures are not informative and are not clearly delineated from the main text of the MS. It would be useful to divide this figure into several figures and place these figures in the appropriate sections.

Response 6 :Thank you for your comments. To enhance the clarity and coherence of our manuscript, we have revised Figures 3 and 4. Figure 3 has been significantly updated in light of structural and content-related revisions, and it now focuses on the dual roles of neutrophils in both promoting and suppressing tumor development. Its corresponding legend has also been revised. Additionally, we have made minor adjustments to Figure 4, adding a descriptive title: The role of neutrophils in tumor metastasis. Lastly, both Figure 3 and Figure 4 have been repositioned to more suitable locations within the manuscript to improve overall clarity and readability.

Comment 7: Minor 2. It was interesting and useful to discuss the interaction of neutrophils with other granulocyte subpopulations (eosinophils and basophils) in this article. Unfortunately, the role of this interaction in both tomologenesis and antitumor defense is still grossly underreported.

Response 7 :Thank you for your helpful comment regarding. In response to your valuable suggestion, we conducted a comprehensive review of recent literature on the roles of eosinophils and basophils in tumor biology. Our findings indeed confirm the existence of dual functional roles played by eosinophils in tumorigenesis, which we have summarized accordingly. However, most of these studies focused on the intrinsic functional mechanisms of eosinophils and lacked investigation into their potential interactions with neutrophils. For this reason, we decided not to incorporate this content into our current manuscript. Regrettably, no substantial body of research was identified that specifically investigates the role of basophils within the tumor microenvironment, highlighting a significant gap in current knowledge. As you correctly noted, the roles of both eosinophils and basophils in tumor development, along with their potential interactions with neutrophils, remain poorly understood and insufficiently explored. Nonetheless, such investigations hold critical importance for advancing our understanding of tumor immunology, and we anticipate that they may yield groundbreaking insights in the near future.

The following section presents a synthesis of the current understanding of eosinophil function in tumor biology, based on the literature we examined:

“Eosinophils play a dual role within the tumor microenvironment: they can suppress tumor growth by enhancing anti-cancer immunity, yet may also contribute to tumor progression under specific circumstances. In suppressing tumor growth, eosinophils activate Th1 and CD8+ T cell responses via the GM-CSF–IRF5 signaling axis, thereby inhibiting intestinal tumor development (https://doi.org/10.1084/jem.20190706). They also facilitate tumor rejection through normalization of tumor vasculature and enhancing CD8+ T cell infiltration into the tumor microenvironment (DOI: 10.1038/ni.3159). In esophageal squamous cell carcinoma, elevated eosinophil infiltration and post-chemoradiotherapy increases in peripheral blood eosinophil counts correlate with improved overall and progression-free survival, indicating their favorable prognostic value (DOI: 10.1097/MD.0000000000024328). Furthermore, through their roles in intestinal immune regulation and host defense, eosinophils indirectly help establish an anti-tumor microenvironment (DOI: 10.1038/s41586-022-05628-7). Conversely, in promoting tumor progression, tumor-associated tissue-resident eosinophils (TATE) facilitate tumor progression in head and neck squamous cell carcinoma through pro-angiogenic activity and metastatic dissemination (https://doi.org/10.1016/j.neo.2022.100855). Overall, the functional impact of eosinophils in tumorigenesis and tumor development varies depending on the tumor type and microenvironmental context, underscoring the necessity for further mechanistic studies to elucidate their precise mechanisms and clinical relevance.

While both neutrophils and eosinophils are key players in the tumor microenvironment, current literature has predominantly focused on the individual functions of each cell type. Few studies have directly explored how interactions between these two cell populations influence tumor development and progression, highlighting a significant knowledge gap in this area. This gap underscores the importance of investigating potential crosstalk between these two myeloid cell populations in cancer biology.”

Reviewer 5 Report

Comments and Suggestions for Authors

Pengcheng Li et. al., reviewed comprehensive role of neutrophil-mediated mechanisms in the tumor microenvironment and highlighted emerging strategies for neutrophil-targeted cancer therapy. However, some more important content should be added.

  • Authors should include current understanding about the recruitment and activity of TANs.
  • Authors should include a table to show the correlation between NETs and progression in different cancer types.
  • Authors should provide the detail role of different immune cells.
  • Authors should provide preclinical and clinical evidence target on different solid tumor and its treatment.
  • More patient outcomes data or real-world evidence is recommended.
  • Authors should include the advantages and disadvantages/side effects of different treatment regimens which is in clinics.
  • Authors should discuss in detail about the challenges in Neutrophil based therapy for the treatment of cancer.

Author Response

Comment 1:Authors should include current understanding about the recruitment and activity of TANs.

Response 1:Thank you very much.Your suggestions were very helpful to us. We have added content regarding the recruitment and activity of tumor-associated neutrophils (TANs), marked with underlines in the manuscript.
lines 499-522: The recruitment of TANs is a critical event in tumor progression, and targeting this process offers various strategies for cancer therapy. Chemokines and their receptors play an essential role in the migration of neutrophils to tumor sites. For example, CXCL2 promotes neutrophil recruitment and inflammatory responses via the CXCR2 axis, and inhibiting CXCR2 can alleviate myeloid inflammation and reverse treatment resistance in prostate cancer [193,194]. The ALX/FPR2 receptor regulates neutrophil infiltration based on ligands such as serum amyloid A, and targeting this receptor holds promise for controlling tumor-associated inflammation [195]. CCL11, a multifunctional chemokine, has a dual role in cancer, demonstrating potential in immune therapy [196]. CXCL5 has been also identified as a key chemokine driving the infiltration of mature pro-tumor neutrophils into lung cancer tissues [197]. Nevertheless, precisely blocking these chemokines without affecting systemic immune function remains a challenge. Additionally, sialic acid-modified nanoplatforms can interfere with the physical or molecular interactions between neutrophils and the tumor microenvironment, blocking their infiltration and enhancing the efficacy of checkpoint blockade therapies [198]. Another strategy involves the direct clearance of pro-tumor myeloid cells, such as PMN-MDSCs, using near-infrared immunotherapy with Ly6G antibodies to selectively eliminate neutrophils in the tumor bed, significantly inhibiting tumor growth and enhancing host immune responses [199,200]. Further studies have shown that pathogens in the tumor microenvironment, such as Fusobacterium nucleatum, recruit TANs by activating the IL-17/NF-κB/RelB pathway, promoting gastric cancer progression, suggesting that targeting the microbiome may indirectly modulate neutrophil behavior [201]. These strategies provide diverse pathways for precisely intervening in neutrophil recruitment.”

Comment 2:Authors should include a table to show the correlation between NETs and progression in different cancer types.

Response 2: Thank you for your comments. Based on your suggestion, we have added a Table 3 in the article detailing the role of NETs in the progression of different cancer types.

" lines 615: Table 3. The role of NETs in different cancer types

lines 616-620: Table 3. The role of NETs in different cancer types. NETs promote tumor progression by enhancing tumor cell proliferation, migration, invasion, angiogenesis, and metastasis, including liver and peritoneal metastasis. They facilitate immune escape, EMT, and pre-metastatic niche formation. High NETs levels correlate with poor prognosis, chemotherapy resistance, and worse survival."

Comment 3:Authors should provide the detail role of different immune cells.

Response 3: Thank you for your helpful comment regarding. Based on your suggestion, we have added content regarding the interactions between neutrophils and other immune cells in the article, marked with underlines.

line274-278 : Importantly, research has demonstrated that MMP9 released by bone marrow-derived neutrophils collaborates with other hematopoietic lineage cells, possibly by modulating the tumor microenvironment or extracellular matrix remodeling, to promote the initiation of squamous cell carcinoma [133].

line280-273 : Within their extracellular fibrous network, NETs capture circulating tumor cells (CTCs), shielding them from the cytotoxic effects of CD8+ T cells and NK cells while simultaneously facilitating metastatic adhesion [118,134,135].

line296-300: in colorectal cancer, N2-polarized neutrophils promote immune evasion by activating latent TGFβ via MMP9, thereby suppressing the function of tumor-infiltrating T cells [142]; whereas in ovarian cancer, these TANs mediate IL-8-dependent immunosuppression (suppressing CD8+ T cells) through the Jagged2 signaling pathway [143].

line312-314: Moreover, N1-polarized neutrophils potentiate anti-tumor immune responses and improve therapeutic outcomes via oxidative stress-mediated suppression of IL-17+ γδ T cells [148].”

Comment 4:Authors should provide preclinical and clinical evidence target on different solid tumor and its treatment.

Response 4: Thank you for your helpful suggestions. Our review of available literature indicates that the majority of neutrophil-targeted anti-cancer therapies remain at the preclinical stage. To date, no clinical trials or direct therapeutic applications involving neutrophils themselves have been reported. However, the neutrophil-to-lymphocyte ratio (NLR) has emerged as a widely accepted prognostic biomarker in oncology. These preclinical findings and NLR-associated evidence are summarized in Chapter 8.

lines 495- 613: 8.Targeting Neutrophils in Cancer Therapy: Strategies and Approaches

TANs play a pivotal role in tumor progression. This section explores various strategies to target and inhibit TANs recruitment, as well as methods for monitoring and modulating neutrophil behavior to improve cancer treatment outcomes.

8.1 Strategies to inhibit TANs recruitment

The recruitment of TANs is a critical event in tumor progression, and targeting this process offers various strategies for cancer therapy. Chemokines and their receptors play an essential role in the migration of neutrophils to tumor sites. For example, CXCL2 promotes neutrophil recruitment and inflammatory responses via the CXCR2 axis, and inhibiting CXCR2 can alleviate myeloid inflammation and reverse treatment resistance in prostate cancer [193,194]. The ALX/FPR2 receptor regulates neutrophil infiltration based on ligands such as serum amyloid A, and targeting this receptor holds promise for controlling tumor-associated inflammation [195]. CCL11, a multifunctional chemokine, has a dual role in cancer, demonstrating potential in immune therapy [196]. CXCL5 has been also identified as a key chemokine driving the infiltration of mature pro-tumor neutrophils into lung cancer tissues [197]. Nevertheless, precisely blocking these chemokines without affecting systemic immune function remains a challenge. Additionally, sialic acid-modified nanoplatforms can interfere with the physical or molecular interactions between neutrophils and the tumor microenvironment, blocking their infiltration and enhancing the efficacy of checkpoint blockade therapies [198]. Another strategy involves the direct clearance of pro-tumor myeloid cells, such as PMN-MDSCs, using near-infrared immunotherapy with Ly6G antibodies to selectively eliminate neutrophils in the tumor bed, significantly inhibiting tumor growth and enhancing host immune responses [199,200]. Further studies have shown that pathogens in the tumor microenvironment, such as Fusobacterium nucleatum, recruit TANs by activating the IL-17/NF-κB/RelB pathway, promoting gastric cancer progression, suggesting that targeting the microbiome may indirectly modulate neutrophil behavior [201]. These strategies provide diverse pathways for precisely intervening in neutrophil recruitment.

8.2 Regulation of neutrophil-derived cytokine release

Neutrophils, as critical effector cells in inflammation and immune regulation, can synthesize and secrete various cytokines, including pro-inflammatory cytokines (TNF-α, IL-1β), chemokines (IL-8, IP-10, MIP-1α), and angiogenic factors (VEGF). These factors not only promote inflammation and immune responses within the tumor microenvironment but also accelerate tumor progression [202]. Therefore, inhibiting the secretion of these cytokines by neutrophils is considered a potential strategy for immunotherapy. TGF-β1 activates the SMAD3 and ERK1/2 signaling pathways, inducing neutrophils in the tumor microenvironment to express tumor-promoting factors, such as OSM and VEGFA mRNA, thereby converting neutrophils into a pro-tumor phenotype [203]. Targeting the TGF-β1 signaling pathway can inhibit the pro-tumor functions of neutrophils, offering a new therapeutic target for precise modulation of the tumor microenvironment.

8.3 Bispecific antibodies enhance neutrophil-mediated antitumor activity

The novel bispecific antibody (TrisomAb) effectively recruits neutrophils as effector cells, enhancing their cytotoxicity against tumor cells. Studies have shown that, in colorectal cancer patients, neutrophils, when exposed to anti-EGFR TrisomAb, are able to efficiently eliminate tumor cells [204].

8.4 Innovative carriers for tumor-targeted nanodrug delivery

Neutrophils, as core effector cells of innate immunity, have become ideal carriers for targeted nanodrug delivery in brain tumors due to their rapid inflammatory response, strong chemotaxis, and ability to cross the blood-brain barrier (BBB) [205–207]. These cells actively participate in inflammation by forming NETs and releasing cytokines, providing a new mechanism for precise drug delivery [205]. In conditions of neuroinflammation or pathological states, the integrity of the BBB is compromised, which facilitates neutrophil migration into the central nervous system (CNS) [208,209]. Acting as "Trojan horses," neutrophils can carry nanodrugs and use their pro-inflammatory properties to deliver anti-cancer drugs precisely to residual glioma cells after surgery, significantly inhibiting tumor recurrence and extending mouse survival [210]. Biomimetic nanocarriers, designed to mimic the surface characteristics of neutrophils, possess immune evasion and targeting capabilities, effectively overcoming the limitations of the BBB and the tumor microenvironment, thereby enhancing drug accumulation and therapeutic efficacy. In a breast cancer model, these carriers demonstrated potential in modulating the microenvironment and inhibiting metastasis, offering new insights into brain tumor treatment [211,212]. Additionally, neutrophil-derived exosomes (NEs-Exos/DOX) leverage their strong inflammatory chemotaxis and BBB penetration ability to successfully deliver doxorubicin to gliomas, significantly inhibiting tumor growth, extending survival, and reducing systemic toxicity [213]. In a breast cancer model, local inflammation induced the recruitment of neutrophils and the release of NETs, further enhancing anti-tumor effects. At the same time, neutrophil infusion helped alleviate systemic inflammation, ensuring clinical safety [214]. These studies collectively highlight the unique advantages of neutrophils and their derived carriers in overcoming the BBB barrier and enhancing targeted brain tumor therapy, providing innovative strategies for precision treatment and suggesting substantial clinical potential.

8.5 Neutrophil-Based Combination Therapy

8.5.1 Combination with immune checkpoint inhibitors: In a glioma mouse model, neutrophil depletion combined with anti-PD-1 antibody treatment significantly inhibited glioma growth and extended mouse survival [215]. In a non-small cell lung cancer (NSCLC) mouse model, CXCL5 gene knockout completely prevented neutrophil accumulation in the lung tissue, which in turn promoted the expansion and cytolytic function of tumor-specific CD8+ T cells [216]. Additionally, studies have shown that the synergistic action of T cells and neutrophils can enhance the efficacy of immunotherapy [217].

8.5.2 Combination with chemotherapy: In pancreatic cancer patients, combining anti-Ly6G therapy to deplete neutrophils with gemcitabine/paclitaxel chemotherapy not only significantly reduced tumor burden and metastatic growth but also enhanced the functionality of tumor-infiltrating CD8+ T cells, effectively suppressed the polarization of CAF, and inhibited chemotherapy resistance via the IL-6/STAT-3 signaling pathway[218].

8.5.3 Other combination strategies with nNeutrophil modulators: Pancreatic melatonin significantly enhanced anti-tumor immune responses by modulating tumor-associated neutrophil infiltration and NETosis in pancreatic adenocarcinoma [219]. Furthermore, mathematical models have been used to optimize the regulation of N1-N2 neutrophil dynamics with TGF-β inhibitors and IFN-β, further enhancing N1 recruitment and effectively inhibiting tumor progression [220]. Probiotics have also been proposed to potentially exert therapeutic effects by inhibiting neutrophil involvement in cancer metastasis [221].

8.6 Neutrophil-Lymphocyte Ratio (NLR): a critical biomarker for cancer prognosis evaluation

Although clinical treatments targeting neutrophils are still in the exploratory stage, several studies in preclinical models and early clinical observations suggest that neutrophil-targeted therapies may improve patient prognosis. Research has shown that an increase in peripheral blood neutrophil count or NLR is an independent predictor of poor prognosis in various cancers. For example, in colorectal cancer patients, elevated neutrophil count is associated with a shorter overall survival (OS) [222]. In glioma patients, the presence of CTCs post-surgery is closely linked to neutrophil-mediated inflammatory immune environments and correlates with poor prognosis [223]. In NSCLC patients, persistent neutrophil elevation and a high NLR (≥5) are associated with negative prognostic value [224]. Additionally, in gastric cancer patients receiving PD-1 antibody treatment, high NLR is also linked to adverse outcomes [225]. These findings highlight the critical role of neutrophils in cancer progression and suggest their potential as therapeutic targets.

In pancreatic cancer patients, dynamic changes in NLR before and during chemotherapy are closely associated with pathological response, disease-free survival (DFS), and OS [218]. Some studies have also proposed that a TICO regimen, consisting of tadalafil, isotretinoin, colchicine, and Omega-3 fatty acids, may inhibit cancer growth and metastasis by reducing high NLR [226]. Although clinical data for this regimen still require validation through large-scale trials, its theoretical foundation and the safety of the drugs involved provide promising possibilities for future clinical applications.

Neutrophils and related indicators, such as NLR, not only serve as prognostic biomarkers but may also guide personalized treatment. Monitoring neutrophils in the tumor microenvironment is crucial for tumor treatment and prognosis assessment. Advances in molecular imaging technologies now enable non-invasive visualization of neutrophil biological behavior, offering strong support for personalized cancer diagnostics and therapeutic strategies [227].”

Comment 5:More patient outcomes data or real-world evidence is recommended.

Response 5: Thank you for your helpful comment regarding.Unfortunately, we were unable to locate any clinical data on the therapeutic use of neutrophils. However, the neutrophil-to-lymphocyte ratio (NLR) is widely recognized in clinical practice as a prognostic marker after tumor therapy. We have included a concise overview of NLR and its clinical relevance. 

line587-613: 8.6 Neutrophil-Lymphocyte Ratio (NLR): a critical biomarker for cancer prognosis evaluation

Although clinical treatments targeting neutrophils are still in the exploratory stage, several studies in preclinical models and early clinical observations suggest that neutrophil-targeted therapies may improve patient prognosis. Research has shown that an increase in peripheral blood neutrophil count or NLR is an independent predictor of poor prognosis in various cancers. For example, in colorectal cancer patients, elevated neutrophil count is associated with a shorter overall survival (OS) [222]. In glioma patients, the presence of CTCs post-surgery is closely linked to neutrophil-mediated inflammatory immune environments and correlates with poor prognosis [223]. In NSCLC patients, persistent neutrophil elevation and a high NLR (≥5) are associated with negative prognostic value [224]. Additionally, in gastric cancer patients receiving PD-1 antibody treatment, high NLR is also linked to adverse outcomes [225]. These findings highlight the critical role of neutrophils in cancer progression and suggest their potential as therapeutic targets.

In pancreatic cancer patients, dynamic changes in NLR before and during chemotherapy are closely associated with pathological response, disease-free survival (DFS), and OS [218]. Some studies have also proposed that a TICO regimen, consisting of tadalafil, isotretinoin, colchicine, and Omega-3 fatty acids, may inhibit cancer growth and metastasis by reducing high NLR [226]. Although clinical data for this regimen still require validation through large-scale trials, its theoretical foundation and the safety of the drugs involved provide promising possibilities for future clinical applications.

Neutrophils and related indicators, such as NLR, not only serve as prognostic biomarkers but may also guide personalized treatment. Monitoring neutrophils in the tumor microenvironment is crucial for tumor treatment and prognosis assessment. Advances in molecular imaging technologies now enable non-invasive visualization of neutrophil biological behavior, offering strong support for personalized cancer diagnostics and therapeutic strategies [227].”

Comment 6:Authors should include the advantages and disadvantages/side effects of different treatment regimens which is in clinics.

Response 6: Thank you for your comments. Despite growing preclinical evidence, clinical studies exploring neutrophil-targeted therapies in cancer treatment remain limited, and no standardized therapeutic protocols have yet been incorporated into clinical practice. Consequently, a comprehensive evaluation of their potential benefits and drawbacks remains challenging at this stage. Nonetheless, accumulating evidence suggests that neutrophil-targeted therapies may emerge as a promising and effective approach in oncology within the foreseeable future.

Comment 7:Authors should discuss in detail about the challenges in Neutrophil based therapy for the treatment of cancer.

Response 7: Thank you for your helpful suggestions. Based on your suggestion, we have detailed the numerous challenges faced by neutrophil research in tumors so far in Chapter 9, and we have also provided some solutions to address these challenges.

line621-680: 9.Conclusions and Future Perspectives

In conclusion, neutrophils, with their multifaceted biological functions and remarkable adaptability, profoundly influence every stage of tumor initiation, progression, and metastasis. They serve as key drivers of chronic inflammation and the formation of pro-metastatic microenvironments, while also possessing the capacity to transform into anti-tumor immune effector cells under specific conditions. This complex bidirectional role and functional heterogeneity establish neutrophils as central regulators within the tumor microenvironment, offering broad prospects for the development of innovative anti-tumor therapies targeting neutrophils.

However, current research faces several limitations and challenges: 1. Incomplete Mechanistic Understanding: Although neutrophils are known to play critical roles in tumor progression, immunosuppression, and metastasis, the underlying mechanisms remain incompletely elucidated. For instance, the precise interactions between neutrophils, tumor cells, and other immune cells—as well as their specific functions at different stages of tumor development—require further investigation. 2. Heterogeneity: Neutrophils exhibit significant heterogeneity across different tumor types, individual patients, and stages of tumor development. This variability complicates research and hinders the development of universal therapeutic strategies. 3. Limited Therapeutic Targets: Current neutrophil-targeted therapeutic approaches are limited, with efficacy varying by tumor type and patient. For example, therapies aimed at blocking neutrophil recruitment are effective in some cases but may be ineffective or cause adverse effects in others. 4. Insufficient Clinical Translation: Despite significant advances in basic research, translating these findings into clinical applications remains challenging. Key issues include accurately assessing neutrophil states and functions and leveraging them as therapeutic targets or biomarkers. 5. Technical Limitations: Existing research methods and technologies face constraints in dissecting neutrophil functions. While single-cell sequencing and spatial omics provide high-resolution data, they are costly and analytically complex. In vivo imaging techniques also lack precision in tracking dynamic neutrophil changes within the TME.

To address these shortcomings, future research on tumor-associated neutrophils will focus on the following directions to overcome current bottlenecks and advance clinical applications: 1. In-Depth Mechanistic Studies: Advanced technologies, such as single-cell sequencing, CRISPR-Cas9 gene editing, and high-throughput screening, will be employed to investigate the mechanistic roles of neutrophils in the TME. Key areas of focus will include their interaction networks with tumor cells and immune cells (e.g., T cells, macrophages), as well as their specific contributions to immune evasion, angiogenesis, and metastasis, aiming to uncover their critical functions in tumor development. 2. Heterogeneity Analysis: Through single-cell sequencing, spatial omics, and multi-omics integration, systematic studies will characterize neutrophil heterogeneity across tumor types, patient populations, and tumor stages. The objectives are to identify commonalities and differences among neutrophil subsets and provide a scientific basis for personalized and targeted therapies. 3. Novel Therapeutic Targets: Leveraging high-throughput screening, bioinformatics, and artificial intelligence-driven drug design, new therapeutic targets will be identified, and innovative therapies developed. These include antibodies targeting neutrophil surface receptors, cytokines or small-molecule drugs modulating neutrophil functions, and neutrophil-based immunotherapies—all aimed at enhancing treatment efficacy and specificity. 4. Clinical Translation: Efforts will focus on bridging basic research and clinical applications by developing neutrophil-related biomarkers for early tumor diagnosis (e.g., liquid biopsy techniques detecting neutrophil phenotypes or secretions), prognosis prediction (e.g., using neutrophil-associated gene expression profiles), and treatment monitoring (e.g., assessing changes in neutrophil states during therapy)—thereby facilitating the translation of research findings into clinical practice. 5. Technological Innovation: Advances in technology will be pursued to develop cost-effective and efficient tools. These include improved single-cell sequencing and spatial omics platforms to reduce costs and simplify data analysis, as well as high-resolution in vivo imaging techniques (e.g., nanoprobes or live imaging) to precisely track neutrophil dynamics in the TME, providing more accurate tools for studying neutrophil functions and behaviors.

These research directions aim to deepen the understanding of neutrophil roles in cancer, address current challenges, and pave the way for innovative diagnostic and therapeutic strategies in oncology.”

Round 2

Reviewer 4 Report

Comments and Suggestions for Authors

The authors have significantly improved the MS, making it more informative and interesting. There are still some remarks to the structure and design of the MS.

  1. It would be appropriate to notice the words “Tumor Progression” to “Tumor Growth Regulation” in the title.

  2. In the abstract, it would be relevant to mention the role of neutorphils in activation of dormant tumor cells. This phentomenon is extremely important for tumor progression.

  3. Section 2: “The ordinary life of Neutrophils” and Section 3 “Bone Marrow and Circulating Neutrophils” can be combined into one section with corresponding subsections. The text describing the effects of TME on neutrophils (lines 112-129 and 202-110) can be moved to later sections on neutrophil effects on tumor.

  4. Subsection 5.2.3, “Pharmacological Enhancement of Tumor Suppression,” appears inappropriate and understated given the prior data. This subsection could logically complement Section 8 on the possibilities of therapeutic regulation of neutrophil antitumor activity.

  5. The title Table 1 “Mechanisms underlying the role of neutrophils in tumor progression” would be better as “Mechanisms underlying the role of neutrophils in tumor growth regulation”.

  6. The captions for Tables 1 and 2, which are below the tables, do not carry additional information. These captions can be removed from the text.

  7. Sections 5 and 6 could be combined into one division with clear subsections and sub-subsections.

  8. Section 9 “Conclusions and Future Perspectives” is difficult to understand, and in need of significant reduction.

Author Response

Comment 1: It would be appropriate to notice the words “Tumor Progression” to “Tumor Growth Regulation” in the title.

Response 1: Thank you very much. Following your suggestion, we have changed the article title to "Neutrophil Spatiotemporal Regulatory Networks: Dual Roles in Tumor Growth Regulation and Metastasis".

Comment 2: In the abstract, it would be relevant to mention the role of neutorphils in activation of dormant tumor cells. This phentomenon is extremely important for tumor progression.

Response 2:Thank you for your comments. Based on your feedback, we have included content about neutrophils activating dormant tumor cells in the abstract.
Lines 28-29: (3) reactivating dormant tumor cells in response to chronic inflammation, viral infection, or stress hormones.”

Comment 3: Section 2: “The ordinary life of Neutrophils” and Section 3 “Bone Marrow and Circulating Neutrophils” can be combined into one section with corresponding subsections. The text describing the effects of TME on neutrophils (lines 112-129 and 202-110) can be moved to later sections on neutrophil effects on tumor.

Response 3:Thank you for your helpful comment. Based on your recommendations, we have merged Section 2 and 3, reordered and renamed the subsections, and logically incorporated the content from lines 112-129 and 202-210 into Section 4.1.1.3.
“2.Neutrophil Biology: Development, Homeostasis, and Circulation

2.1 Neutrophil Development and Functional Maturation

2.2 Bone Marrow Reservoir and Regulatory Functions

2.3 Circulatory Dynamics and Homeostatic Regulation

Lines 278-286: Furthermore, tumor-derived small extracellular vesicles (sEVs) deliver serine protease inhibitors (serpins), which reprogram neutrophils toward a pro-tumorigenic phenotype characterized by an extended lifespan and an immunosuppressive expression pattern. These neutrophils functionally enhance tumor cell epithelial-mesenchymal transition and potently inhibit cytotoxic CD8+ T cells, thereby reducing their tumor-killing efficacy [130]. In the tumor microenvironment, IL-1β induces G-CSF production, promoting both neutrophil expansion and functional polarization. Tumor-educated neutrophils gain the capacity to suppress CD8+ T cell cytotoxicity, thereby facilitating tumor progression[131–135].”

Comment 4: Subsection 5.2.3, “Pharmacological Enhancement of Tumor Suppression,” appears inappropriate and understated given the prior data. This subsection could logically complement Section 8 on the possibilities of therapeutic regulation of neutrophil antitumor activity.

Response 4:Thank you for your helpful suggestions.Based on your feedback, we have incorporated the original content of Subsection 5.2.3 into the current Subsection 6.5.3 in a more logical manner.

Lines 556-570:The role of neutrophils in cancer immunotherapy has been significantly enhanced through various pharmacological and biological interventions. Pharmacological interventions such as angiotensin-converting enzyme inhibitors (ACEIs) and angiotensin II type 1 receptor (AGTR1) antagonists can drive neutrophil polarization toward the antitumor N1 phenotype, thereby enhancing their cytotoxic potential [142]. Clinically, increased intratumoral infiltration of N1 neutrophils has been correlated with improved patient outcomes. For example, suppression of the TGF-β signaling pathway has been shown to reprogram neutrophils into the N1 phenotype, leading to effective tumor suppression [143–150]. Computational modeling has further refined the regulation of N1/N2 neutrophil balance by TGF-β inhibitors and IFN-β, favoring N1 recruitment and significantly impeding tumor progression [211]. Moreover, melatonin has been found to enhance antitumor immunity through modulation of tumor-associated neutrophil infiltration and NETosis in pancreatic ductal adenocarcinoma [212]. Probiotics have also been implicated in exerting therapeutic benefits through the inhibition of neutrophil-mediated cancer metastasis [213].”

Comment 5: The title Table 1 “Mechanisms underlying the role of neutrophils in tumor progression” would be better as “Mechanisms underlying the role of neutrophils in tumor growth regulation”.

Response 5: Thank you very much.We have changed the title of Table 1 to "Table 1 Mechanisms underlying the role of neutrophils in tumor growth regulation".

Comment 6: The captions for Tables 1 and 2, which are below the tables, do not carry additional information. These captions can be removed from the text.

Response 6: Thank you for your comments. We have removed the captions for Tables 1 and 2 as suggested.

Comment 7: Sections 5 and 6 could be combined into one division with clear subsections and sub-subsections.

Response 7: Thank you for your valuable suggestion. Based on your recommendation, we have combined Sections 5 and 6 into a single division, with clearly organized subsections and sub-subsections. Additionally, we have reordered and renamed the corresponding subsections and sub-subsections to enhance clarity and coherence.

Comment 8: Section 9 “Conclusions and Future Perspectives” is difficult to understand, and in need of significant reduction.

Response 8:Thank you for your helpful comment regarding. Based on your recommendations, we have revised the "Conclusions and Future Perspectives" section, significantly reducing its content to make it clearer and more concise.
"Lines 605-633: In summary, the functional heterogeneity and plasticity of neutrophils play a pivotal role in tumor initiation, progression, and metastasis. These cells exhibit a dual role: either promoting inflammatory responses and tumor metastasis, or acquiring anti-tumorigenic phenotypes under specific conditions. This functional duality positions neutrophils as critical regulators within the tumor microenvironment, highlighting their potential as targets for innovative anti-cancer therapeutic strategies.

However, current research faces several limitations and challenges: 1.         Unclear Mechanisms: the intricate molecular mechanisms underlying the interactions among neutrophils, tumor cells, and other immune components, along with their evolving roles during cancer progression, are not yet fully elucidated. 2. High Variability: neutrophil behavior displays significant heterogeneity across different cancer types, individual patients, and tumor stages. This variability complicates mechanistic studies and limits the development of broadly applicable therapeutic strategies. 3. Technical Limitations: current methodologies, including single-cell sequencing and advanced imaging technologies, are often expensive, technically challenging, and may lack standardization. Accurately characterizing neutrophil functional states and identifying tumor-specific subpopulations remains a major technical barrier.

To address these limitations, future research should focus on the following areas:1. In-depth Mechanistic Studies: Utilizing cutting-edge approaches such as gene editing (e.g., CRISPR-Cas9) and multi-omics profiling to dissect the precise and context-dependent interactions between neutrophils, tumor cells, and immune cells. These efforts aim to uncover both shared and cancer-specific molecular mechanisms. 2.Clinical Translation: Building upon deeper mechanistic understanding, integrating artificial intelligence and big data analytics to identify novel diagnostic and therapeutic targets. This will facilitate the development of personalized precision medicine strategies for early cancer detection, diagnosis, and monitoring of treatment responses. 3. Technological Advancements: Advancing the development of cost-effective, user-friendly tools—such as high-throughput sequencing platforms and high-resolution imaging systems—to improve accessibility of affordable diagnostics and alleviate the clinical burden of cancer."